# The influence of TC0668 on glycometabolism modulation in *Chlamydia muridarum*-infected host cells

Nanyan Yu,[1,2] Xuan Chen,[1] Wenjing Yang,[1] Yang Zhou,[1] Yuchen Hu,[1] Zhou Zhou[1]

**ABSTRACT** *Chlamydia,* an obligate intracellular parasite, depends entirely on host cells for energy and biosynthesis while exerting pathogenic effects through its virulence factors. *Chlamydia muridarum* (*Cm*), an alternative model strain to *Chlamydia trachomatis* (*Ct*), modulates cellular metabolism to enhance its survival and pathogenicity during infection. We found that TC0668, a crucial *Cm* virulence protein associated with fallopian tube lesions in infected mice, induces a hypermetabolic state in host cells upon *Cm* stimulation. This results in alterations in glucose consumption, mitochondrial TCA cycle activity, aerobic glycolysis, and intracellular ATP levels. Specifically, infection with the *Cm* TC0668[wt] strain in HeLa and HUVEC cells led to the activation of PI3K (p110) and substantial phosphorylation of AKT at S473. This activation was significantly reduced by LY-294002, a potent PI3K/AKT pathway inhibitor, which led to decreased glucose consumption and ATP levels in HUVECs. However, in HeLa cells, inhibition of the pathway primarily affected GLUT1 expression and ATP levels without impacting glucose consumption. These findings underscore the pivotal role of PI3K/AKT signaling in regulating cellular glycometabolism under the influence of the TC0668 protein during *Cm* infection.

**IMPORTANCE** Previous studies have identified that TC0668, as a virulence factor involved in the formation of fallopian tube hydrosalpinx caused by *Chlamydia muridarum* (*Cm*), is primarily involved in metabolic processes, cellular processes, and biological regulation, and there are notable differences in PI3K activation and AKT phosphorylation induced by *Cm tc0668* single-gene strains. However, the relationship between TC0668's influence on *Cm*-regulated glycometabolism and the activation of the PI3K/AKT pathway remains unclear. Our study established a vitro cell infection model of *Cm* using HeLa cells and HUVEC cells, and employed techniques such as Western blotting to reveal a novel mechanism of TC0668 in enhancing the pathogenicity of *Cm* by regulating host glycometabolism. The study advances our understanding of intracellular pathogen-host interactions and provides novel therapeutic strategies for *Chlamydia* infections.

**KEYWORDS** Chlamydia muridarum, TC0668, glycometabolism, PI3K/AKT, infection

Address correspondence to Zhou Zhou, susiezhou99503@163.com.

Nanyan Yu, Xuan Chen, and Wenjing Yang contributed equally to this article. The author order was determined by workload.

Yang Zhou and Yuchen Hu contributed equally to this article.

The authors declare no conflict of interest.

See the funding table on p. 14.

*C*hlamydia trachomatis (*Ct*) is a major pathogen responsible for sexually transmitted diseases (STDs) in humans. Infectious elementary bodies (EBs) of *Ct* invade host cells, form vesicle-like structures, and differentiate into reticulate bodies (RBs), which proliferate through binary fission (1, 2). In the later stages of infection, RBs revert to EBs. Following the rupture of both the mature inclusion body membrane and the host cell membrane, numerous EBs are released to infect neighboring cells (3). As an obligate intracellular bacterium, *Chlamydia* depends on host cells for energy and nutrients during infection, while its virulence factors, such as those affecting cellular metabolism, drive pathogenicity (4–7). Understanding the metabolic changes caused by *Ct* infection is

crucial for gaining insights into its pathogenic mechanisms and could lead to novel approaches to prevent and control *Ct* infections.

During normal cellular metabolism, host cells uptake glucose from the extracellular environment via glucose transporters (GLUTs), converting it into pyruvate via glycolysis. Pyruvate then enters the mitochondria, undergoing oxidative decarboxylation to form coenzyme A (CoA), which fuels the tricarboxylic acid (TCA) cycle, producing nicotinamide adenine dinucleotide (NADH) and reduced flavin adenine dinucleotide (FADH2). To support ATP synthesis and provide energy for physiological activities, NADH drives the transmembrane proton gradient through the NADH-CoQ oxidoreductase complex (complex I), cytochrome C reductase (complex III), and cytochrome C oxidase (complex IV). Similarly, FADH2 contributes to the proton gradient through the succinate-CoQ oxidoreductase complex (complex II), complex III, and complex IV (8). In pathogen-infected host cells, the expressions of glucose transporters (GLUTs) are upregulated to increase glucose uptake, which is then metabolized into substantial amounts of pyruvate. Pyruvate is either utilized in mitochondrial oxidative phosphorylation (OXPHOS) via the TCA cycle and electron transport chain (ETC) to release energy and nutrients or converted into lactate and additional energy through the aerobic glycolytic pathway, facilitated by lactate dehydrogenase (4, 9–14).

In a comprehensive study involving genome-wide RNA interference screening and metabolic profiling, Marion Rother et al. highlighted that *Ct* infection shifts host cells toward aerobic glycolysis, positively regulating energy and nucleotide metabolism (9). Additionally, this infectious process enhances RNA synthesis to fulfill the energetic and biosynthetic demands necessary for *Chlamydia* growth and development. Siegl et al. demonstrated that *Ct* infection induces AKT phosphorylation through PI3K activation, influencing host cell metabolism via MDM2-p53 interaction (15). Some of these findings have also been partially corroborated in studies of *Chlamydia muridarum* (*Cm*), which serves as an effective alternative model to *Ct*. *Cm* similarly enhances pathogenicity by modulating the metabolism of infected cells (16). However, the specific components of *Chlamydia* responsible for regulating cellular metabolism and affecting its pathogenicity remain unknown.

Our research identified TC0668 as a key pathogenic component of *Cm*. Mutations in the tc0668 gene significantly reduced *Cm*'s ability to induce tubal lesions in mice, indicating its role in pathogenicity. Specifically, the TC0668 G216* mutation results from a substitution of T with G at position 216 in the TC0668 gene, converting a glycine-encoding codon into a stop codon, which leads to the premature termination of peptide synthesis. TC0668 is a virulence factor affecting the upper genital tract and potentially causing hydrosalpinx, although its exact pathogenic mechanism remains unclear. Initially, we infected HeLa cells *in vitro* with *Cm* single-gene differential strains and conducted proteomic analysis. The results indicated that the differentially expressed proteins were primarily involved in metabolic processes, cellular processes, and biological regulation (15). Subsequent research confirmed that *Cm tc0668* single-gene strains induced PI3K activation and AKT phosphorylation in markedly different ways (17). This supports the idea that, similar to *Ct*, *Cm* may activate the PI3K/AKT signaling pathway and modulate the MDM2-p53 axis to regulate host cell metabolism.

Carbohydrates, as prime carbon sources and metabolic byproducts of glucose, can be converted into various carbon-containing compounds, participating actively in multiple metabolic pathways and biological functions. Glucose metabolism, in particular, is critical to the infection processes of numerous pathogens. Accordingly, this study further explored the effect of TC0668 on *Cm* glycometabolism in host cells and its underlying pathogenic mechanisms. Specifically, it aims to investigate the relationship between TC0668's influence on *Cm*-regulated glycometabolism and the activation of the PI3K/AKT pathway.

## MATERIALS AND METHODS

### Antibodies

Antibodies against PI3K (#4249), AKT (#4691), phospho-Akt (#4060), OGDH (#26865), and COX IV (#4850) were obtained from Cell Signaling. Glut1 (ab115730) was sourced from Abcam. Beta-actin (Cat No. 66009-1-Ig) was purchased from Proteintech. Horse-radish peroxidase (HRP)-labeled goat anti-rabbit secondary antibody (SA00001-2) and HRP-labeled goat anti-mouse secondary antibody (SA00001-1) were also provided by Proteintech.

### Cell culture

Human cervical carcinoma epithelial cells (HeLa) were cultured at 37°C in 5% CO2 using Dulbecco's modified Eagle's medium (DMEM; Sigma) supplemented with 10% fetal calf serum (FCS; Excell Bio). Human umbilical vein endothelial cells (HUVECs) were grown in Endothelial Cell Medium (ECM; ScienCell) with 10% FCS under the same conditions. The use of these cells was approved by the local ethics committee. Cells were infected with or without the PI3K inhibitor LY294002 (Abcam) at a final concentration of 20 µM. HeLa cells were exposed to $Cm$ TC0668$^{wt}$ or TC0668$^{mut}$ monoclonal strains at a multiplicity of infection (MOI) of 1.0, followed by centrifugation at 1,000 rpm for 1 hour at 37°C. After removal of the inoculum, the cells were replenished with fresh DMEM with 10% FCS for 6, 12, 18, or 24 hours. HUVECs were similarly infected with these strains at an MOI of 1.0, followed by centrifugation at 1,000 rpm for 1 hour at 37°C. After removal of the inoculum, the cells were replenished with fresh ECM with 10% FCS for 12, 24, 36, or 42 hours.

### Preparation of *Chlamydia muridarum* strain

The $Cm$ TC0668 mutation-type monoclonal strain (TC0668$^{mut}$) features a nonsense mutation at codon 216 [Glycine], resulting in the TC0668 G216* [TC0668*, Short] variant. This mutation leads to premature termination of the protein. $Cm$ TC0668$^{wt}$ and TC0668$^{mut}$ strains are differentiated by the presence or absence of the tc0668 gene and its function. HeLa cells were incubated with $Cm$ TC0668$^{wt}$ and TC0668$^{mut}$ strains for 18–24 hours, then harvested, and lysed using precooled SPG buffer on ice. The resulting cell suspension was centrifuged at 4°C, 3,000 rpm for 10 minutes. The supernatant was collected and stored at −80°C. Fresh strains were used for each experiment.

### Western blotting

Infected and uninfected control HeLa cells and HUVECs were lysed on ice for 30 minutes using RIPA buffer supplemented with 1 mM PMSF, 1% protease inhibitor, and 1% phosphatase inhibitor. The lysates were sonicated at 4°C and then centrifuged at 12,000 rpm for 10 minutes to obtain the supernatant. The concentration of parvalbumin was measured using the BCA method. The supernatant was mixed with SDS loading buffer at a 4:1 ratio and boiled at 100°C for 10 minutes to denature the proteins. Proteins were separated by 12.5% SDS-PAGE, transferred to a PVDF membrane, and blocked at room temperature for 2 hours with 5% skim milk in TBS containing 0.1% Tween-20 and then shaken gently. The membrane was incubated overnight at 4°C with a diluted primary antibody. After five washes with TBST, it was incubated with HRP-conjugated goat anti-rabbit or anti-mouse secondary antibody at 37°C for 1 hour. Following five additional washes with TBST, protein bands were detected using a chemiluminescent imaging system with Immunolo ECL Ultra Western HRP Substrate. Control samples (uninfected groups) were processed synchronously with infected groups and harvested at the same endpoint of the incubation period. Grayscale analysis of Western blot images was performed using ImageJ software, and values of target protein and loading control (β-actin) bands were measured. The relative expression level was calculated as the ratio of the target protein to loading control (Target Grayscale/β-actin Grayscale). Error

bars represent the standard deviation (SD) derived from three independent biological replicates.

## Analysis of cellular glucose uptake and lactate production rates

Cell culture medium from infected or uninfected control groups was collected, and the supernatant was obtained by centrifugation at 4°C, 3,000 rpm for 10 minutes. Glucose and lactate levels in the supernatant were measured using an Automatic Biochemistry Analyzer (HITACHI 7600 series) in the biochemical laboratory of the Department of Clinical Laboratory at the First Affiliated Hospital of University of South China. The analyzer was calibrated with reference substances before each use. Control samples (uninfected groups) were processed synchronously with infected groups and harvested at the same endpoint of the incubation period.

## Determination of intracellular ATP levels

The infected or uninfected control cells were lysed on ice with ATP detection lysate for 15 min, collected into a 1.5 mL centrifuge tube, and centrifuged at 4°C, 12,000 g for 5 min to obtain supernatants. The working solution for ATP detection was prepared, and then it was added into a 96-well enzyme-labeled plate protected from light according to the addition of 100 µL/well and placed at room temperature for 5 min. After the background ATP was completely consumed, 20 µL supernatants were respectively added into the detection wells, which were then mixed rapidly with a pipette and placed at room temperature for 15 min. The RLU value was measured with a fluorescence microplate reader (Bio-Tek). The ATP concentration of the sample was calculated according to the standard curve. The protein was homogenized by BCA, and the final ATP value (nmol/mg protein) for the sample was calculated. Control samples (uninfected groups) were processed synchronously with infected groups and harvested at the same endpoint of the incubation period.

## Statistical analysis

Grayscale scanning was performed using ImageJ software, and mapping and statistical analysis were conducted with GraphPad Prism 8.0 software. All data were acquired from three independent experiments. The data were analyzed using two-way ANOVA and one-way ANOVA. Error bars represent the standard deviation (SD) derived from three independent biological replicates. Statistical significance was determined based on a significance level of 0.05.

## RESULT

### TC0668 affected cellular glycometabolism

An escalating body of research has revealed that *Chlamydia* not only exhausts energy and nutrients from host cells but also actively regulates cellular metabolism to ensure its optimal infection. The major energy source for cell metabolism is glucose. Increased glucose uptake is a common consequence of *Chlamydia* infection, as observed in cells infected with *Ct* and *Chlamydia psittaci* (*Cps*) (13, 18). In our study, glucose levels in the supernatants of TC0668^wt- and TC0668^mut-infected cells gradually decreased over time (Fig. 1A and B). Notably, compared to the TC0668^mut-infected group, glucose levels in the culture medium dropped more dramatically following infection with the *Cm* TC0668^wt strain, particularly in HeLa cells at 12 and 18 h, and in HUVECs at 24, 36, and 42 h (Fig. 1A and B). Furthermore, GLUT1 expression was significantly upregulated in the *Cm* TC0668^wt-infected group, in HeLa cells at 18 and 24 h and in HUVECs at 42 h (Fig. 1C and D). The deficiency of TC0668 impairs *Cm*'s ability to regulate glucose uptake in host cells.

Once glucose enters the cell, it is converted into pyruvate in the cytoplasm through several pathways. Pyruvate then travels to the mitochondria, where it undergoes oxidative decarboxylation before entering the TCA cycle and contributing to the electron transport chain. We observed that in HeLa cells and HUVECs infected with the *Cm*

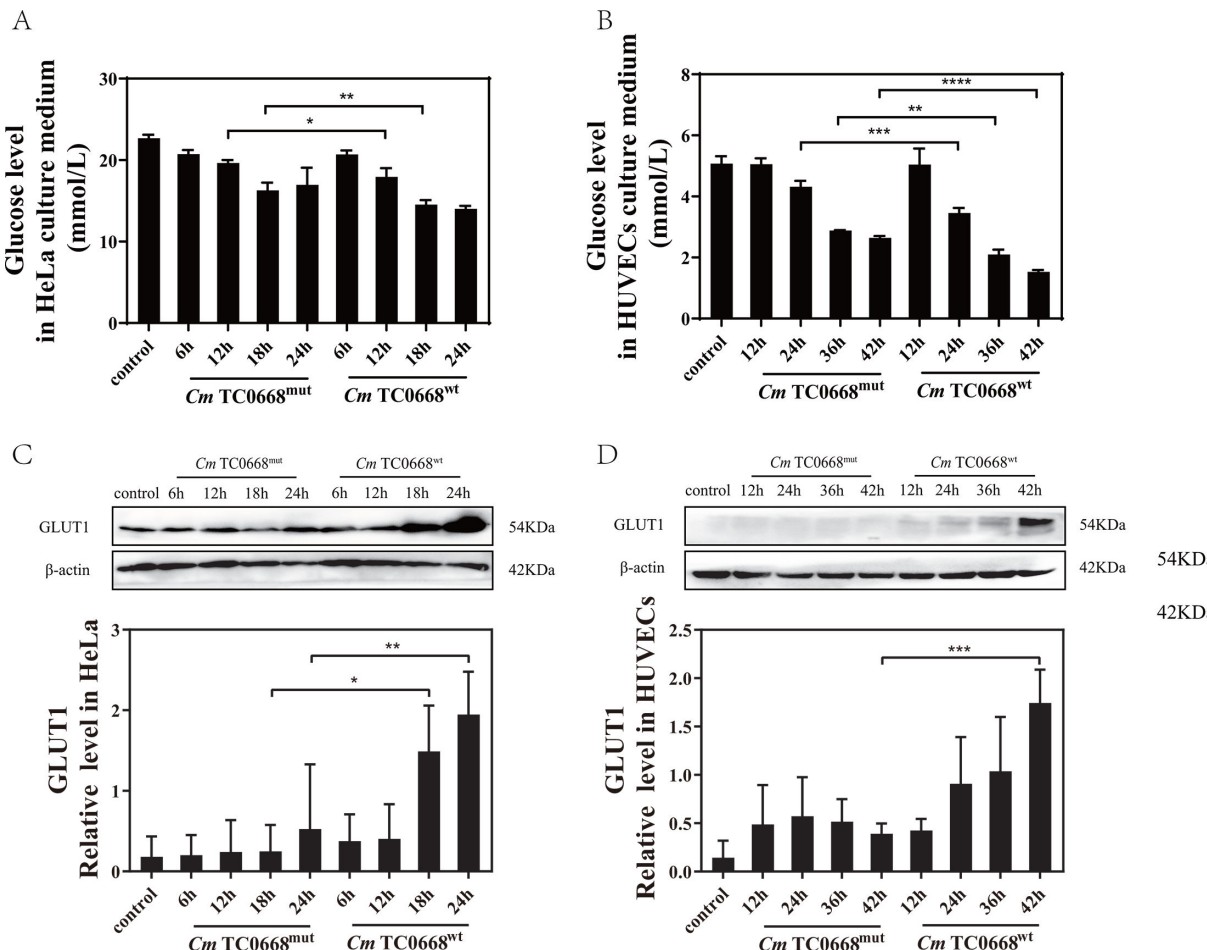

**FIG 1** Deficiency in TC0668 impairs glucose uptake regulated by *Cm* in host cells (two-way ANOVA, *$P < 0.05$, **$P < 0.01$, ***$P < 0.001$, and ****$P < 0.0001$). (A) Glucose levels in the supernatant of HeLa cells infected with *Cm* TC0668[wt] and TC0668[mut] strains were measured using the ACA assay at 6, 12, 18, and 24 hours post-infection. (B) Similarly, glucose concentrations in the supernatant of HUVECs infected with these strains were analyzed at 12, 24, 36, and 42 hours using ACA. (C) GLUT1 protein expression was assessed by Western blotting in HeLa cells infected with *Cm* TC0668[wt] and TC0668[mut] strains at 6, 12, 18, and 24 hours. (D) Western blotting was also used to evaluate GLUT1 expression in HUVECs infected with these strains at 12, 24, 36, and 42 hours. β-actin served as a loading control.

TC0668[wt] strain, the TCA cycle-related protein OGDH increased progressively in the early stages of infection but decreased in the later stages (Fig. 2A and B). The *Cm* TC0668[wt] strain was more effective at upregulating OGDH expression compared to the *Cm* TC0668[mut] strain, with notable effects seen after 12 hours of infection in HeLa cells and 24 hours in HUVECs (Fig. 2A and B). The absence of TC0668 likely impairs *Cm*'s ability to regulate the mitochondrial TCA cycle in host cells. In contrast, COX IV, a protein involved in the electron transport chain, showed an initial increase followed by a decrease during infection with both strains. There was no significant difference in COX IV expression between the *Cm* TC0668[wt] and TC0668[mut] strains throughout the infection cycle (Fig. 2A and B). This suggests that TC0668 does not play a role in *Cm*'s regulation of COX IV in host cells.

When mitochondria are damaged and pyruvate cannot enter them efficiently for OXPHOS, or when cells are under stress from pathogen invasion, pyruvate is converted into lactate in the cytoplasm via LDH, producing ATP to sustain high metabolic activity. We found that lactate concentrations in the culture supernatant increased progressively with infection time in both *Cm* TC0668[wt]- and TC0668[mut]-infected HeLa cells and HUVECs. Notably, lactate levels rose more significantly in cells infected with

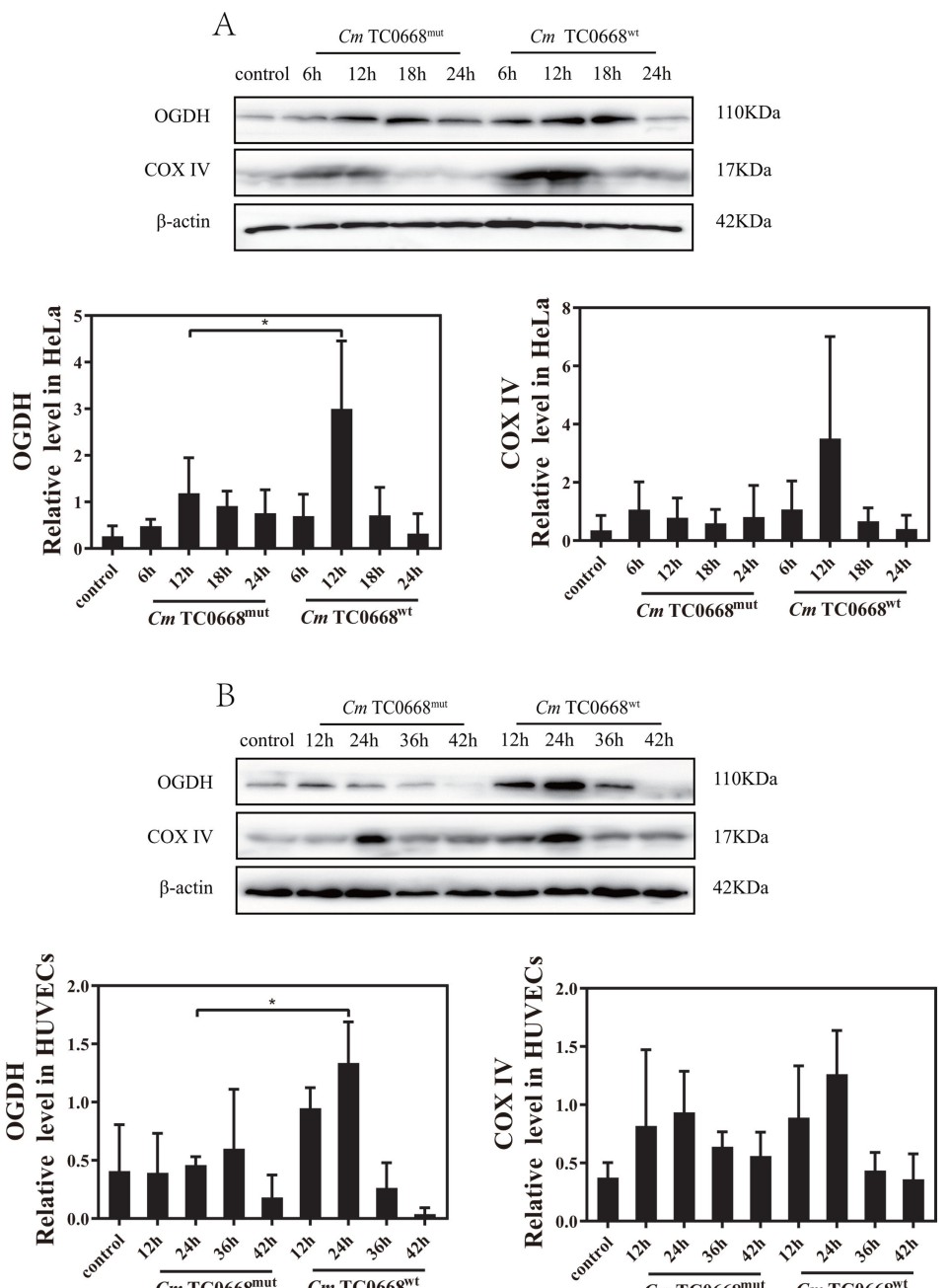

**FIG 2** The effect of Cm TC0668 on mitochondrial oxidative phosphorylation (OXPHOS) in host cells was evaluated (two-way ANOVA, *$P < 0.05$). Western blot analysis was conducted to measure the expression levels of OGDH and COX IV in HeLa cells infected with Cm TC0668wt and TC0668mut strains for various durations (6, 12, 18, and 24 hours) (A), and in HUVECs infected for different periods (12, 24, 36, and 42 hours) (B). β-actin served as a loading control.

*Cm* TC0668wt, particularly in HeLa cells after 18–24 hours and in HUVECs after 24–42 hours (Fig. 3A and B). These findings suggest that the *Cm* TC0668wt strain induces greater glucose consumption and lactate production in host cells compared to the *Cm* TC0668mut strain, indicating that TC0668 affects *Cm*-regulated aerobic glycolysis in host cells.

ATP and various nutrients are produced through mitochondrial OXPHOS and aerobic glycolysis. Previous studies have shown that intracellular ATP levels fluctuate dynamically during *Chlamydia* growth and development. Our data reveal that in HeLa cells infected with *Cm* TC0668wt and TC0668mut strains, intracellular ATP initially increased before

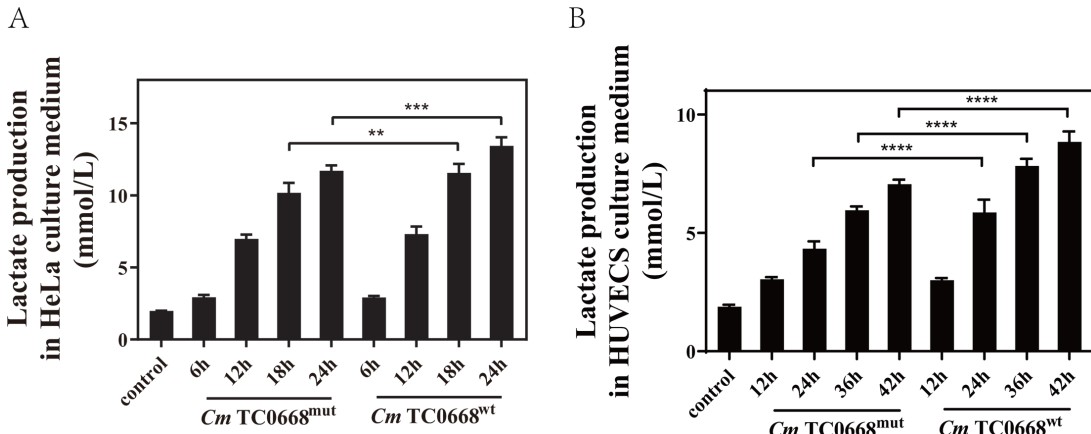

**FIG 3** TC0668 deficiency disrupts *Cm*-regulated aerobic glycolysis in host cells (two-way ANOVA, **$P < 0.01$, ***$P < 0.001$, and ****$P < 0.0001$). Supernatants were collected after infecting HeLa cells with *Cm* TC0668wt and TC0668mut strains for 6, 12, 18, and 24 hours (A) and HUVECs for 12, 24, 36, and 42 hours (B). Lactate levels in the supernatants were analyzed using ACA.

decreasing (Fig. 4A). In *Cm* TC0668wt-infected HUVECs, ATP rose during the early stages but declined later, while in *Cm* TC0668mut-infected HUVECs, ATP increased initially but plateaued in the later stages (Fig. 4B). Overall, ATP levels were elevated early on and then decreased in the TC0668wt-infected groups compared to the TC0668mut-infected group, suggesting that TC0668 absence alters *Cm*'s regulation of intracellular ATP levels.

In summary, the removal of TC0668 significantly impacts *Cm*'s regulation of glucose uptake, the mitochondrial TCA cycle, aerobic glycolysis, and intracellular ATP levels in host cells.

## The activation of the PI3K/AKT signaling pathway was enhanced by the TC0668 protein of *Cm*

We investigated the impact of TC0668 on the activation of the PI3K/AKT signaling pathway by monitoring PI3K activation and the serine phosphorylation of AKT at Ser473. The results showed that PI3K (p110) activation and AKT phosphorylation were significantly higher in HeLa cells and HUVECs infected with the *Cm* TC0668wt strain compared to those infected with the *Cm* TC0668mut strain (Fig. 5A and B). This indicates that the *Cm* TC0668wt strain is more effective in inducing PI3K activation and

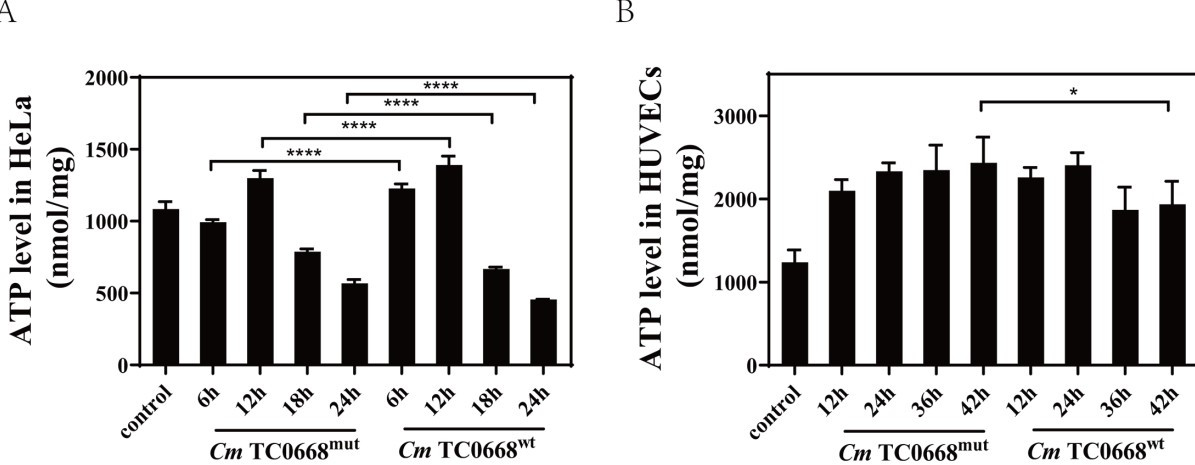

**FIG 4** The absence of TC0668 affects *Cm*'s modulation of intracellular ATP levels in host cells (two-way ANOVA, *$P < 0.05$; ****$P < 0.0001$). ATP changes were measured using fluorescence microplate analyzers in HeLa cells at 6, 12, 18, and 24 hours (A) and in HUVECs at 12, 24, 36, and 42 hours (B) following infection with *Cm* TC0668wt and TC0668mut strains.

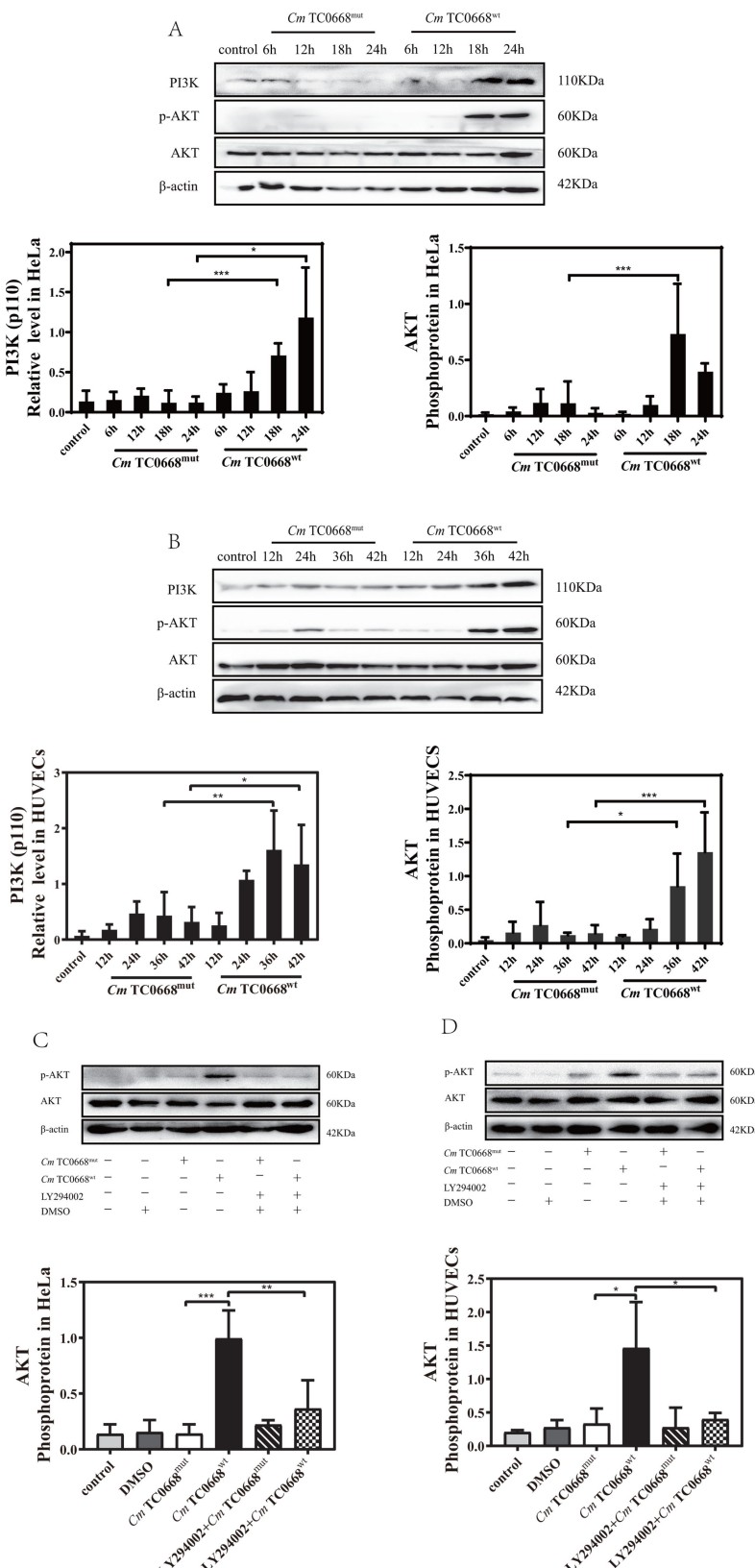

**FIG 5** The TC0668 compound derived from *Cm* significantly stimulates the activation of the PI3K/AKT signaling pathway. (A) Western blot analysis was performed to assess the expression of PI3K and the phosphorylation of AKT in HeLa cells infected with the *Cm* TC0668$^{wt}$ and TC0668$^{mut}$ strains at 6, 12, 18,

Fig 5 (Continued)

and 24 hours (two-way ANOVA, *P < 0.05; ***P < 0.001). (B) The presence of PI3K and p-AKT in HUVECs was examined by Western blotting following infection with the *Cm* TC0668<sup>wt</sup> and TC0668<sup>mut</sup> strains at 12, 24, 36, and 42 hours (two-way ANOVA, *P < 0.05, **P < 0.01, and ***P < 0.001) (C) HeLa cells were treated with the PI3K inhibitor LY294002 at a concentration of 20 µM and subsequently infected with the *Cm* TC0668<sup>wt</sup> and TC0668<sup>mut</sup> strains for 18 hours (one-way ANOVA, *P < 0.05, **P < 0.01, and ***P < 0.001). (D) The phosphorylation of AKT in HUVECs infected with the *Cm* TC0668<sup>wt</sup> and TC0668<sup>mut</sup> strains was assessed after treatment with a specific PI3K inhibitor at a concentration of 20 µM for 42 hours (one-way ANOVA, *P < 0.05, **P < 0.01, and ***P < 0.001). β-actin was used as a loading control.

AKT phosphorylation, suggesting that TC0668 enhances *Cm*'s ability to activate PI3K and promote AKT phosphorylation. Additionally, LY294002, a potent PI3K inhibitor that blocks AKT phosphorylation and inhibits the PI3K-AKT pathway, reduced AKT phosphorylation in HeLa and HUVECs infected with the *Cm* TC0668<sup>wt</sup> strain but not in those infected with the TC0668<sup>mut</sup> strain (Fig. 5C and D). This confirms that LY294002 inhibits PI3K/AKT signaling activated by the *Cm* TC0668<sup>wt</sup> strain, thereby validating that the presence of TC0668 affects the activation of the PI3K/AKT signaling pathway by *Cm*.

## The role of the PI3K/AKT pathway in the influence of TC0668 on cellular glycometabolism

We have demonstrated that the inhibitor exerts no effect on the growth or development of either chlamydial strain (Fig. S2). To examine the potential influence of the PI3K/AKT signaling on the effect of TC0668 on *Cm*-regulated glycometabolism, we analyzed changes in glucose and lactate levels in the culture medium supernatant, as well as intracellular GLUT1 and ATP levels, in HeLa cells infected with the *Cm* TC0668<sup>wt</sup> and TC0668<sup>mut</sup> strains for 18 h and in HUVECs infected for 42 h. Prior to infection, the cells were pretreated with 20 µM LY294002, a specific PI3K inhibitor, for 1 h. We observed no significant difference in glucose levels between *Cm* TC0668<sup>wt</sup>- and TC0668<sup>mut</sup>-infected HeLa cells, with or without the specific PI3K inhibitor (Fig. 6A). In HUVECs treated with the PI3K inhibitor, the *Cm* TC0668<sup>mut</sup>-infected group also showed no noticeable changes in glucose levels, whereas glucose levels in the supernatant of *Cm* TC0668<sup>wt</sup>-infected cells significantly increased (Fig. 6B). These results suggest that the effect of TC0668 on *Cm*-regulated glucose consumption in HUVECs is linked to PI3K/AKT pathway activation. Western blot analysis further revealed a marked reduction in GLUT1 levels in *Cm* TC0668<sup>wt</sup>-infected HeLa cells and HUVECs following suppression of the PI3K/AKT signaling pathway (Fig. 6C and D), indicating that inhibiting this pathway decreases GLUT1 expression. However, lactate levels in the culture medium of *Cm* TC0668<sup>wt</sup>- and *Cm* TC0668<sup>mut</sup>-infected HeLa cells and HUVECs showed no significant change after PI3K inhibitor pretreatment (Fig. 7A and B). This suggests that inhibition of the PI3K/AKT signaling pathway does not significantly impact TC0668's role in *Cm*-regulated lactate production. Additionally, inhibition of the PI3K/AKT pathway resulted in a notable increase in intracellular ATP levels in *Cm* TC0668<sup>wt</sup>-infected HeLa cells, but not in the *Cm* TC0668<sup>mut</sup>-infected group (Fig. 8A). In contrast, PI3K/AKT pathway inhibition led to a significant rise in intracellular ATP levels in both *Cm* TC0668<sup>wt</sup>- and TC0668<sup>mut</sup>-infected HUVECs (Fig. 8B).

## DISCUSSION

As an obligate intracellular bacterium, *Ct* actively modulates cellular metabolism to meet its energy and biosynthesis needs for growth and development, thereby ensuring effective infection and pathogenicity. Our previous research identified TC0668 as a virulence factor involved in the formation of fallopian tube hydrosalpinx caused by *Cm*, a strain related to *Ct*. However, the exact pathogenic mechanism remains unclear (19). HeLa and HUVEC cells have been widely used in chlamydial studies (15, 20, 21). Most importantly, previous studies have confirmed that both murine cells BM12.4 and

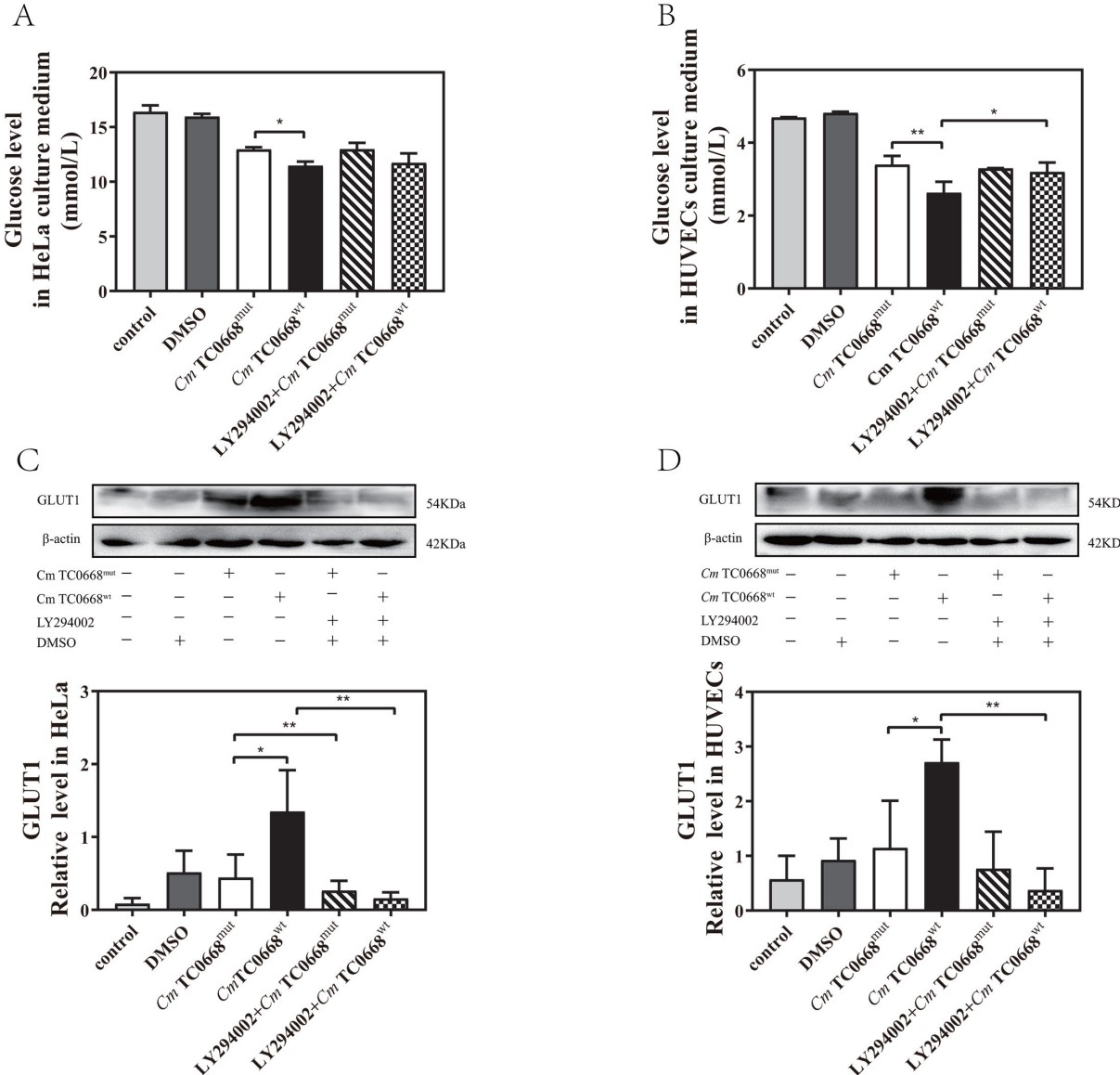

**FIG 6** Regulation of the PI3K/AKT pathway by TC0668 of *Cm* affects glucose consumption (one-way ANOVA, *P < 0.05; **P < 0.01). (A and C) HeLa cells were pretreated with 20 µM LY294002, a PI3K inhibitor, for 1 hour, and then infected with *Cm* TC0668^wt and TC0668^mut strains for 18 hours. Glucose levels in the culture supernatant were quantified using the ACA assay, and GLUT1 expression was assessed by Western blotting. (B and D) HUVECs were treated with 20 µM LY294002 and subsequently infected with *Cm* TC0668^wt and TC0668^mut strains for 42 hours. GLUT1 levels were analyzed by Western blotting, with β-actin as the loading control.

human cells infected with different chlamydial strains (*C. trachomatis* serovars A to H, L1 to L3, *C. muridarum*, *C. pneumoniae,* and *C. caviae*) exhibited the same protein tyrosine phosphorylation patterns (22). In addition, PI3K/AKT signaling pathway and glucose metabolic regulatory networks are highly conserved in mammals (23, 24). It is practical to apply human HeLa and HUVEC cells to investigate the role and mechanism of TC0668 in regulating cellular glucose metabolism. This work aims to provide a foundation for understanding the role of TC0668 in the pathogenic mechanisms of *Cm*.

Previous studies have shown that *Chlamydia* infection significantly increases glucose consumption in mouse fibroblasts (18). In line with these findings, we observed a marked increase in glucose consumption in the culture supernatant of HeLa cells and HUVECs infected with the *Cm* TC0668^wt strain compared to those infected with the *Cm* TC0668^mut strain. This suggests that TC0668 may play a role in regulating glucose

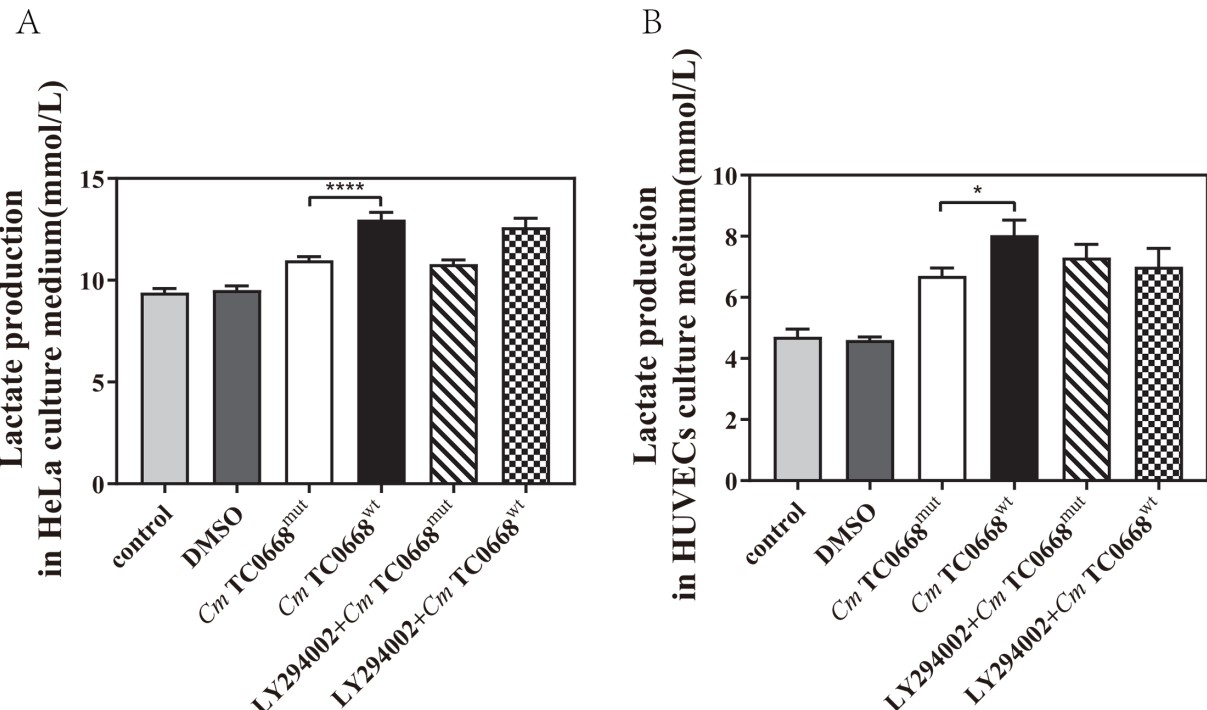

**FIG 7** Regulation of aerobic glycolysis by TC0668 of *Cm* may not rely on PI3K/AKT signaling activation (one-way ANOVA, *P < 0.05; ****P < 0.0001). (A and B) Lactate levels in the culture supernatant were measured using the ACA assay after HeLa cells (A) or HUVECs (B) were infected with *Cm* TC0668wt and TC0668mut strains for 18 hours and 42 hours, respectively, following a 1 hour pretreatment with the LY294002 inhibitor.

consumption by *Chlamydia* in host cells. GLUT1, a key protein responsible for cellular glucose uptake, has been implicated in this process. Ojcius DM et al. demonstrated that infection with *Chlamydia psittaci* (*Cps*) led to increased glucose consumption and upregulation of GLUT1 expression in HeLa cells (18). Consistent with these findings, we observed a progressive increase in GLUT1 expression in HeLa cells and HUVECs infected with the *Cm* TC0668wt strain compared to the *Cm* TC0668mut strain. Based on these

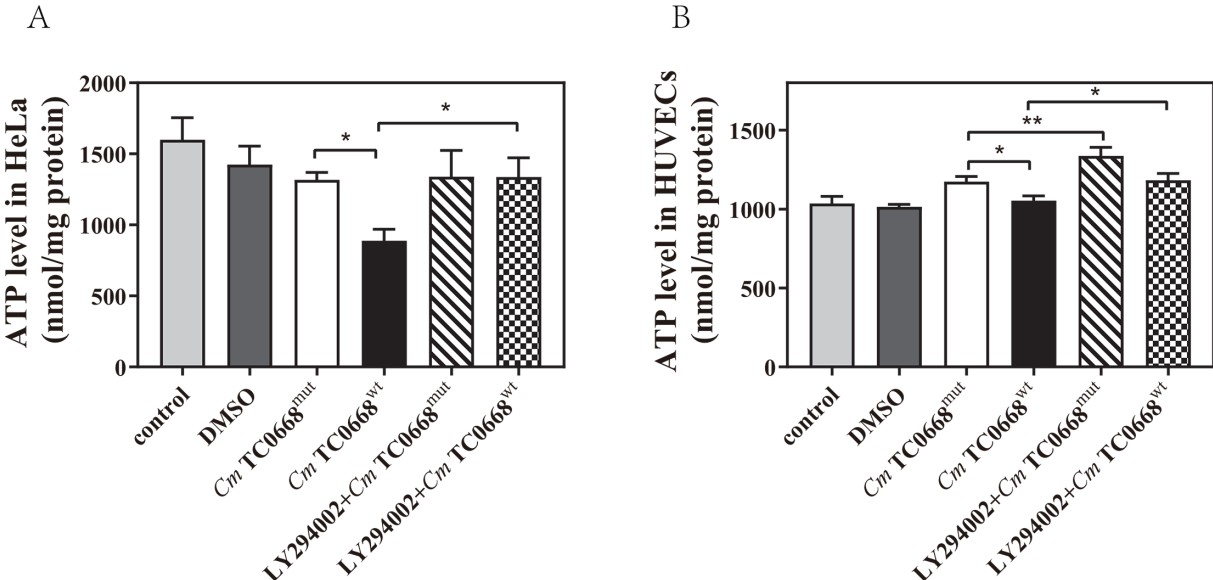

**FIG 8** Modulation of intracellular ATP levels by TC0668 of *Cm* in relation to PI3K/AKT signaling pathway activity (one-way ANOVA, *P < 0.05; **P < 0.01). (A and B) After inhibiting the PI3K/AKT pathway, intracellular ATP levels were measured using a fluorescent microplate reader.

results, we propose that TC0668 in *Cm* may regulate GLUT1 expression in host cells, thereby affecting glucose uptake by the pathogen. TC0668, likely membrane-bound, may regulate host metabolism indirectly. While its enzymatic activity remains uncharacterized, given the important role of deubiquitination in regulating GLUT1 stability (25) and the fact that AKT phosphorylates USP4 to enhance its stability (26), we propose that TC0668 may sustain GLUT1 expression indirectly through the PI3K/AKT-USP4 axis. Specifically, TC0668-induced AKT activation could phosphorylate and stabilize deubiquitinases (e.g., USP4), thereby reducing GLUT1 ubiquitination and subsequent proteasomal degradation.

Glucose, the primary source of cellular energy, is transported into cells where it is metabolized into pyruvate. Pyruvate then undergoes oxidative decarboxylation and enters the mitochondria to participate in the TCA cycle. This process, facilitated by enzymes such as citrate synthase, isocitrate dehydrogenase, and the 2-oxoglutarate dehydrogenase complex (OGDC), produces NADH, FADH2, and ATP. Metabolic analyses by Nadine Vollmuth et al. have demonstrated that *Ct* reprograms the TCA cycle to support pyrimidine biosynthesis (27). Additionally, *Cps* infection in HeLa cells has been shown to downregulate TCA cycle genes, highlighting its impact on host cell metabolism (28). OGDC, a key enzyme complex in the TCA cycle, comprises OGDH, dihydrolipoamide succinyltransferase (DLST), and dihydrolipoamide dehydrogenase (DLD). OGDH, as the E1 subunit of OGDC, is crucial for mitochondrial redox sensing (29). Our study observed a significant upregulation of intracellular OGDH expression during the early stages of infection with the *Cm* TC0668$^{wt}$ strain, followed by a decrease in later stages. This initial increase in OGDH expression likely reflects the strain's ability to induce hypermetabolism and enhance mitochondrial TCA cycle activity. However, prolonged hypermetabolism may result in mitochondrial damage or fragmentation, impairing TCA cycle function. NADH and FADH2 produced by the TCA cycle enter the ETC, where they undergo complete oxidation to generate substantial energy. Cytochrome c oxidase, a critical component of the ETC, is essential for its assembly and function. Specifically, the COX IV subunit is crucial for the proper functioning of cytochrome C oxidase and affects OXPHOS (30). Previous research by Azenabor et al. suggested that Chlamydia pneumoniae (*Cpn*) may enhance cytochrome C oxidase activity in macrophages (31). However, in our study, no significant differences in COX IV expression were observed between *Cm* TC0668$^{mut}$ and TC0668$^{wt}$ strains. This suggests that *Cm* TC0668 may modulate other subunits or proteins involved in the mitochondrial ETC, affecting *Cm*'s control over cellular mitochondrial OXPHOS. Alternatively, TC0668 might have a limited role in regulating mitochondrial electron respiratory chain regulation by *Cm*, warranting further investigation into the specific mechanisms involved.

When mitochondrial function is compromised, the efficient entry of pyruvate into the mitochondria for the TCA cycle and ETC is disrupted, leading to cellular stress or increased vulnerability to pathogens. To support cellular hypermetabolism, pyruvate can be converted to lactate and a small amount of ATP by lactate dehydrogenase in the cytoplasm. Research has shown that *Cps* infection in mouse fibroblasts enhances lactate production (18), and patients with severe psittacosis have elevated lactate dehydrogenase levels (32). Similarly, *Ct* infection in HeLa cells shifts metabolism toward aerobic glycolysis, increasing lactate levels (9). Our study found that *Cm* TC0668$^{wt}$-infected HeLa cells and HUVECs exhibited higher lactate levels and lower glucose levels compared to the *Cm* TC0668$^{mut}$ strain. This supports the idea that TC0668 may influence *Cm*'s regulation of cellular aerobic glycolysis (10).

It is well-established that *Chlamydia* growth and development rely on host cell ATP for energy, with ATP synthesis linked to mitochondrial OXPHOS and glycolysis (8, 11, 33). Under normal aerobic conditions, OXPHOS produces ATP more efficiently than glycolysis, serving as the primary ATP source in mammalian cells. However, tumor cells exhibit the "Warburg effect," producing lactate from glucose even in the presence of oxygen (34, 35). This phenomenon is also observed in proliferating stem cells (34), certain differentiated stem cells (36), and cells stimulated by pathogens (37).

Research indicates that *Chlamydia* infection induces a hypermetabolic state early on, causing mitochondrial elongation and increased ATP production (38), which supports intracellular growth. As the infection progresses, mitochondrial disruption reduces ATP synthesis via OXPHOS, shifting energy production toward aerobic glycolysis as an alternative ATP source. Given that *Chlamydia* actively proliferate and consume significant ATP, intracellular ATP levels fluctuate dynamically throughout the infection (11, 13, 14, 39, 40). Our study confirmed this pattern: after infection with two *Cm* strains, intracellular ATP levels initially increased and then decreased, peaking during the intermediate stage. Cells infected with the *Cm* TC0668[wt] strain showed a more pronounced early-phase increase and a greater late-phase decrease in ATP levels compared to those infected with the *Cm* TC0668[mut] strain, suggesting that TC0668 may influence *Cm*'s regulation of ATP dynamics. Additionally, our previous metabolomics analysis identified differential metabolites associated with glucose, succinate, and lactate in cells infected with both *Cm* TC0668[wt] and TC0668[mut] strains. This implies that TC0668 may affect aerobic glycolysis and mitochondrial OXPHOS in *Cm*-infected cells. In summary, TC0668 appears to impact host cell regulation of glucose consumption, mitochondrial TCA cycle activity, aerobic glycolysis, and intracellular ATP levels mediated during *Cm* infection. However, further research is needed to fully elucidate the specific mechanisms underlying cellular glucose metabolism regulation. The PI3K/AKT signaling pathway is integral to various cellular processes, including cell proliferation, migration, apoptosis, lipid and glucose metabolism, regulation of mitochondrial membrane potential, and angiogenesis. It plays a crucial role in maintaining normal cellular functions (41–45), and disruptions in this pathway are associated with altered cell growth and tumor development (46–48). Our study identified significant variations in the expression levels of key molecules involved in the PI3K/AKT pathway activation, such as PI3K and phosphorylated AKT (p110), consistent with previous findings from our research group (17). Melstrom et al. reported that PI3K/AKT pathway upregulation increases GLUT1 expression (49), while Ohnishi et al. found that inhibiting the pathway effectively reduces GLUT1 expression in osteoblasts (50). Supporting these findings, our study demonstrated that inhibiting the PI3K/AKT pathway reduced GLUT1 expression in HeLa cells and HUVECs infected with the *Cm* TC0668[wt] strain. Notably, inhibiting the PI3K/AKT pathway also suppressed glucose consumption in the culture medium of HUVECs infected with the *Cm* TC0668[wt] strain, although this effect was not significant in HeLa cells. These observations suggest that the TC0668 protein modulates GLUT1 expression in both HeLa cells and HUVECs regulated during *Cm* infection. While glucose consumption in HUVECs is associated with PI3K/AKT signaling activation, in HeLa cells, it seems to be independent of this pathway.

The differences between tumor cells and normal cells in morphology, metabolism, and growth patterns may explain these observations. HeLa cells, derived from a tumor, might increase glucose consumption level as a compensatory mechanism to counteract the effects of PI3K inhibitors. Wang et al. clearly demonstrated that GLUT1 and GLUT3 were significantly upregulated in *Chlamydia* infection (25). We focus on GLUT1 due to its highly conserved and widely distributed glucose transporter in mammalian cells (51, 52). Further investigation of other GLUT subtypes, such as GLUT3 in specific contexts, is needed. Our previous research suggested that the TC0668 protein may influence NF-κB activation in HeLa cells during *Cm* infection. Future studies should explore whether compensatory enhancement of NF-κB signaling or alternative pathways might occur following the inhibition of the PI3K/AKT signaling pathway. Additionally, with 14 different GLUT transporters facilitating glucose transport in various tissues, including extensively studied GLUT2, GLUT3, and GLUT4 (53, 54), we hypothesize that other GLUT transporters might be involved in glucose consumption during *Cm* TC0668[wt] strain infection in HeLa cells. Further investigation is necessary to identify the specific underlying factors involved.

While several studies have highlighted the critical role of PI3K/AKT signaling in promoting cellular aerobic glycolysis (48, 55, 56), our findings indicate that inhibiting this pathway does not significantly affect lactate production in either HeLa cells or

HUVECs. This suggests that the impact of the TC0668 protein on *Cm* regulation of aerobic glycolysis may be independent of PI3K/AKT pathway activation. Nevertheless, PI3K inhibitors modestly increased intracellular ATP levels in both cell types, suggesting that the TC0668 protein's effect on *Cm* regulation of aerobic glycolysis might involve alternative pathways, such as p53-related pathways (16, 57). Despite the increase in aerobic glycolysis following stimulation with the *Cm* TC0668^wt strain, mitochondrial OXPHOS remains the primary mode of ATP production, suggesting a minimal role of TC0668 in *Cm*'s regulation of glycolysis. Further validation is necessary to determine the dominant metabolic mode involved.

Cellular metabolic reprogramming is complex, and this study demonstrates that the TC0668 protein induces a hypermetabolic state in *Cm*-infected host cells. This state is characterized by increased GLUT1 expression, elevated glucose consumption, and regulation of mitochondrial TCA cycle activity. Aerobic glycolysis is enhanced, contributing to regulated intracellular ATP levels. Additionally, PI3K/AKT pathway activation is linked to GLUT1 expression and ATP levels. However, the exact mechanisms by which TC0668 regulates host cell metabolism and contributes to *Cm*'s pathogenicity remain unclear. Further research is needed to investigate additional metabolic pathways and signaling mechanisms that TC0668 may affect in host cells during *Cm* infection.

## ACKNOWLEDGMENTS

Z.Z. and N.Y. conceived the study, drafted and supervised the manuscript. X.C., W.Y., Y.Z., and Y.H. helped with the methodology of the study. X.C., W.Y., Y.Z., and Y.H. helped with the writing, review, and editing of the manuscript. All authors have read and agreed to the published version of the manuscript.

This work is supported by the National Natural Science Foundation of China (grant number 82372284 to Z.Z.).

## AUTHOR AFFILIATIONS

[1]Institute of Pathogenic Biology, Hunan Provincial Key Laboratory for Special Pathogens Prevention and Control, Hunan Province Cooperative Innovation Center for Molecular Target New Drug Study, Hengyang Medical School, University of South China, Hengyang, Hunan, China
[2]Department of Laboratory Medicine, Yunnan Provincial Hospital of Traditional Chinese Medicine, Kunming, Yunnan, China

## AUTHOR ORCIDs

Nanyan Yu ⓘ http://orcid.org/0009-0003-2769-193X

## FUNDING

| Funder | Grant(s) | Author(s) |
| --- | --- | --- |
| National Natural Science Foundation of China | 82372284 | Zhou Zhou |

## ADDITIONAL FILES

The following material is available online.

### Supplemental Material

**Figures S1 and S2 (Spectrum03051-24-s0001.docx).** Figure S1: Schematic representation of cellular glucose metabolism, encompassing glycolysis and oxidative phosphorylation. Figure S2: The PI3K inhibitor LY294002 exerts no effect on the growth or development of Cm TC0668^wt and TC0668^mut strains.

## Open Peer Review

**PEER REVIEW HISTORY (review-history.pdf).** An accounting of the reviewer comments and feedback.

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
