## [Reviewer comments · Microbiology Spectrum]

Microbiology Spectrum

The influence of TC0668 on glycometabolism modulation in *Chlamydia muridarum*-infected host Cells

Nanyan Yu, Xuan Chen, Wenjing Yang, Yang Zhou, Yuchen Hu, and Zhou Zhou

Corresponding Author(s): Zhou Zhou, University of South China Institute of Pathogenic Biology

Review Timeline:

Submission Date:	November 25, 2024
Editorial Decision:	March 27, 2025
Revision Received:	May 31, 2025
Editorial Decision:	June 30, 2025
Revision Received:	September 3, 2025
Accepted:	September 12, 2025

Editor: Simone Filardo

Reviewer(s): The reviewers have opted to remain anonymous.

Transaction Report:

DOI: <https://doi.org/10.1128/spectrum.03051-24>

Re: Spectrum03051-24 (The influence of TC0668 on glycometabolism modulation in Chlamydia muridarum-infected host cells)

Dear Ms. Nanyan Yu:

Thank you for the privilege of reviewing your work. Below you will find my comments, instructions from the Spectrum editorial office, and the reviewer comments.

Revision Guidelines

Sincerely,
Simone Filardo
Editor
Microbiology Spectrum

Reviewer #1 (Comments for the Author):

Summary

The authors report a series of experiments linking the presence of a chlamydial virulence protein (TC0668) that is associated with fallopian tube lesions in infected mice, with the induction of a hypermetabolic state in HeLa and Human Umbilical Vein Endothelial cells (HUVECs). The area under investigation is topical, interesting, and significant. The phenotypes the authors'

focus on appear to be solid, however, there are a number of concerns regarding the cell lines / controls chosen for these experiments, and the statistical analyses used to test for significance.

Major Comments:

- Throughout the manuscript, the authors never describe 1- how many technical / biological replicates were used in generating the graphical depictions of their datasets, 2- what their error bars represent, and 3- what statistical tests were used for each dataset depicted. This is a major omission.
- Throughout the manuscript, the authors utilize whole-cell lysates for experimentation, as opposed to utilizing percoll-gradient enriched EBs. There is concern that the phenotypes being investigated (particularly represented by the 6hpi time point for HeLa cells) could be adversely impacted by the exposure of fresh host cells to cell debris present in whole-cell lysates. The appropriate negative control (when using EB lysates) would be lysates prepared from uninfected cells. The authors do not appear to have done this, relying instead on uninfected cells for their negative control(s).
- It is unclear why unpaired t-tests were chosen as opposed to two-way ANOVAs (with multiple comparisons) in assessing significance for the majority of the figures that represent experimentation conducted with 2 independent variables (Cm strain and time). Running experiments for different time durations and only reporting 'significance by unpaired t-test for specific time points gives the impression of presenting biased comparisons.
- For every use of PI3K inhibitors, the authors need to demonstrate that they are not impacting the developmental cycle of their two Cm strains; there is a high likelihood that their observations could be indirect effects.

Minor Comments:

- Please use page and line numbers for all subsequent submissions / revisions
- If using shorthand for *Chlamydia trachomatis* and *Chlamydia muridarum*, please italicize (ie. Ct and Cm)
- Introduction: paragraph 3, line 8: *Chlamydia muridarum*, as opposed to *Chlamydia murine*
- Methods: The authors need to justify the use of human cell lines for their examination, which is primarily examining the impacts of a *Chlamydia muridarum* virulence factor. Put more simply, given that the phenotype leading to this investigation was murine-specific, why did the authors not utilize murine cells for their study? Given that Conrad 2016 used HeLa cells to demonstrate that there was no difference in development between wt and TC0668, it would be nice to confirm this in murine cell lines.
- Methods: Please indicate how cells were infected for each experiment (rocking, static, centrifugation), the duration of initial infections, and whether crude lysates were removed and replaced with fresh culture medium prior to subsequent incubation.
- paragraph 1, lines 3-5: reference(s) for this statement are requested
- Figure 2: The GLUT1 data looks strong, but the Glucose levels look weaker. Please indicate in figure legends exactly what type of statistical analysis was conducted and how many replicates data presented represents.
- Figure 3: From the blot shown in panel A, it certainly looks like COX IV is elevated in the wt post 18hpi. Please describe how the 'relative levels' were calculated, what the error bars represent, and how many experimental replicates were used to generate each data point.
- It seems puzzling that, given the author's conclusion that TC0668 is responsible for inducing cells into a hypermetabolic state, the absence of this state would have no discernable impact on Chlamydial biology / development. How do the authors account for this?
- To the above point, given what we know about the growth of *C. muridarum* in culture, it could be that the authors utilize such 'nutrient-rich' conditions (ie. serum + human cell lines) that any impact on the host cell's metabolic state is immaterial to chlamydial development. This could be addressed with a relatively straightforward approach: remove FCS (not required for Cm culture) from culture medium and examine potential differences in ifu counts from HeLa cells infected with WT vs. TC0668 null strains at early (12-16hpi) and later (24 hpi) developmental time points. Similar experimentation could also be conducted between strains in a murine-derived cell line.

Reviewer #2 (Comments for the Author):

- The authors should consider moving figure 1 to supplemental. Alternatively, they could expand the schematic to integrate their proposed model. As is, it adds little to the paper.

- The mechanism for the observed alterations to host cell metabolism should be discussed. The authors' lab has previously demonstrated that TC0668 is likely Cm membrane-bound rather than an inclusion membrane protein. Do the authors predict that TC0668 has enzymatic activity? Does TC0668 regulate deubiquitination of GLUT1, a known phenomenon (see Wang et al 2017, PMID: 29040458)? While speculative, further discussion of the hypothesized mechanism would enhance the discussion.
- Please indicate the number of biological replicates performed for each assay.
- Please indicate the conditions of the control samples for each assay, especially the temporal experiments. Is the control taken at the end of an identical incubation period as the final timepoint?
- Given the intrinsic variability of western blot densitometry, the authors should provide more detail on the statistical analyses used in each instance. If using ANOVA, what post hoc tests were used?
- The authors should detail how they determine relative levels in the blot quantitation. I assume this is relative to loading control, but explicit detail is needed.
- While not statistically significant by the tests used, the displayed western blot and presented data imply that COXIV is affected by TC0668. The authors would benefit from displaying individual data points as opposed to averages and error bars. The error bars should also be defined (SD, SEM, etc.).
- Regarding the discussion section on the 14 different GLUT transporters, Wang et al 2017 assessed transcription of all 14 and found only GLUT1 and GLUT3 upregulated. Discussion of that paper could help resolve some of those questions.

In the manuscript titled “The influence of TC0668 on glycometabolism modulation in *Chlamydia muridarum*-infected host cells”, Yu et al continue their studies of TC0668, a gene that they and others have previously shown to be important for infection in mouse models. They demonstrated that the presence of wild-type TC0668 increases host cell expression of several genes involved in host cell metabolism. They also showed a modest drop in media glucose concentration that is exacerbated in the presence of TC0668. Wild-type TC0668 also appears to be critical for AKT phosphorylation, particularly during mid- to late-stage infection. Overall, the data appear to support the conclusion that TC0668 plays a role in host metabolism. However, some additional discussion and further clarifications are required to complete this paper.

Comments:

- The authors should consider moving figure 1 to supplemental. Alternatively, they could expand the schematic to integrate their proposed model. As is, it adds little to the paper.
- The mechanism for the observed alterations to host cell metabolism should be discussed. The authors’ lab has previously demonstrated that TC0668 is likely Cm membrane-bound rather than an inclusion membrane protein. Do the authors predict that TC0668 has enzymatic activity? Does TC0668 regulate deubiquitination of GLUT1, a known phenomenon (see Wang et al 2017, PMID: 29040458)? While speculative, further discussion of the hypothesized mechanism would enhance the discussion.
- Please indicate the number of biological replicates performed for each assay.
- Please indicate the conditions of the control samples for each assay, especially the temporal experiments. Is the control taken at the end of an identical incubation period as the final timepoint?
- Given the intrinsic variability of western blot densitometry, the authors should provide more detail on the statistical analyses used in each instance. If using ANOVA, what post hoc tests were used?
- The authors should detail how they determine relative levels in the blot quantitation. I assume this is relative to loading control, but explicit detail is needed.
- While not statistically significant by the tests used, the displayed western blot and presented data imply that COXIV is affected by TC0668. The authors would benefit from displaying individual data points as opposed to averages and error bars. The error bars should also be defined (SD, SEM, etc.).
- Regarding the discussion section on the 14 different GLUT transporters, Wang et al 2017 assessed transcription of all 14 and found only GLUT1 and GLUT3 upregulated. Discussion of that paper could help resolve some of those questions.

Overall, the data support a role for TC0668 in the modulation of host glycometabolism and set the stage for future investigation. The manuscript will benefit from additional clarification of some of the analyses.

Manuscript number: Spectrum03051-24

Title: The influence of TC0668 on glycometabolism modulation in Chlamydia muridarum-infected host cells

Dear Editors and Reviewers:

We thank all the editors and reviewers for their valuable comments and suggestions. We have carefully revised the manuscript to enhance its clarity, and our changes/additions to the manuscript are given in the red text. Our point to point responses are presented in the following. We hope that the revision would satisfactorily address the comments and concerns of the editors and reviewers.

Reply to the reviewer's comments:

Reviewer #1:

Summary

The authors report a series of experiments linking the presence of a chlamydial virulence protein (TC0668) that is associated with fallopian tube lesions in infected mice, with the induction of a hypermetabolic state in HeLa and Human Umbilical Vein Endothelial cells (HUVECs). The area under investigation is topical, interesting, and significant. The phenotypes the authors' focus on appear to be solid, however, there are a number of concerns regarding the cell lines / controls chosen for these experiments, and the statistical analyses used to test for significance.

Reply: Thank you for your affirmation and encouragement. The paper has been updated accordingly and detailed corrections are listed as below:

Major Comments:

- Throughout the manuscript, the authors never describe 1- how many technical / biological replicates were used in generating the graphical depictions of their datasets, - what their error bars represent, and 3- what statistical tests were used for each dataset depicted. This is a major omission.

Reply: Thanks for your helpful suggestions. We have carefully addressed each of your concerns as follow:

- 1) All data were from three independent experiments.
- 2) Error bars represent the standard deviation (SD) derived from three independent biological replicates.
- 3) The data were analyzed using multiple t-tests or one-way ANOVA.
- 4) These details have been added to the revised “Statistical Analysis” or captions.

Detailed modifications of this part are as follow:

Page 8, lines 178-181: All data were acquired from three independent experiments. The data were analyzed using multiple t-tests and one-way ANOVA. Error bars represent the standard deviation (SD) derived from three independent biological replicates.

Page 25, lines 579-587: Figure 1. Deficiency in TC0668 impairs glucose uptake regulated by *Cm* in host cells (multiple t-tests, * $P < 0.05$, ** $P < 0.01$, *** $P < 0.001$). (A) Glucose levels in the supernatant of HeLa cells infected with *Cm* TC0668^{wt} and TC0668^{mut} strains

were measured using the ACA assay at 6, 12, 18, and 24 hours post-infection. (B) Similarly, glucose concentrations in the supernatant of HUVECs infected with these strains were analyzed at 12, 24, 36, and 42 hours using ACA. (C) GLUT1 protein expression was assessed by Western blotting in HeLa cells infected with *Cm* TC0668^{wt} and TC0668^{mut} strains at 6, 12, 18, and 24 hours. (D) Western blotting was also used to evaluate GLUT1 expression in HUVECs infected with these strains at 12, 24, 36, and 42 hours. β -actin served as a loading control.

Page 25-26, lines 588-593: Figure 2. The effect of *Cm* TC0668 on mitochondrial oxidative phosphorylation (OXPHOS) in host cells was evaluated (multiple t-tests, $*P < 0.05$). Western blot analysis was conducted to measure the expression levels of OGDH and COX IV in HeLa cells infected with *Cm* TC0668^{wt} and TC0668^{mut} strains for various durations (6, 12, 18, and 24 hours), and in HUVECs infected for different periods (12, 24, 36, and 42 hours). β -actin served as a loading control.

Page 26, lines 594-598: Figure 3. TC0668 deficiency disrupts *Cm*-regulated aerobic glycolysis in host cells (multiple t-tests, $*P < 0.05$, $**P < 0.01$, $***P < 0.001$). Supernatants were collected after infecting HeLa cells with *Cm* TC0668^{wt} and TC0668^{mut} strains for 6, 12, 18, and 24 hours (A), and HUVECs for 12, 24, 36, and 42 hours (B). Lactate levels in the supernatants were analyzed using ACA.

Page 26, lines 599-603: Figure 4. The absence of TC0668 affects *Cm*'s modulation of intracellular ATP levels in host cells (multiple t-tests, $*P < 0.05$, $**P < 0.01$). ATP changes were measured using fluorescence microplate analyzers in HeLa cells at 6, 12,

18, and 24 hours (A) and in HUVECs at 12, 24, 36, and 42 hours (B) following infection with *Cm* TC0668^{wt} and TC0668^{mut} strains.

Page 26-27, lines 604-616: Figure 5. The TC0668 compound derived from *Cm* significantly stimulates the activation of the PI3K/AKT signaling pathway. (A) Western blot analysis was performed to assess the expression of PI3K and the phosphorylation of AKT in HeLa cells infected with the *Cm* TC0668^{wt} and TC0668^{mut} strains at 6, 12, 18, and 24 hours (multiple t-tests, **P* < 0.05, ***P* < 0.01, ****P* < 0.001). (B) The presence of PI3K and p-AKT in HUVECs was examined by Western blotting following infection with the *Cm* TC0668^{wt} and TC0668^{mut} strains at 12, 24, 36, and 42 hours (multiple t-tests, **P* < 0.05, ***P* < 0.01, ****P* < 0.001) (C) HeLa cells were treated with the PI3K inhibitor LY294002 at a concentration of 20 μM and subsequently infected with the *Cm* TC0668^{wt} and TC0668^{mut} strains for 24 hours (one-way ANOVA, **P* < 0.05, ***P* < 0.01, ****P* < 0.001). (D) The phosphorylation of AKT in HUVECs infected with the *Cm* TC0668^{wt} and TC0668^{mut} strains was assessed after treatment with a specific PI3K inhibitor at a concentration of 20 μM for 42 hours (one-way ANOVA, **P* < 0.05, ***P* < 0.01, ****P* < 0.001). β-actin was used as a loading control.

Page 27, lines 617-624: Figure 6. Regulation of the PI3K/AKT pathway by TC0668 of *Cm* affects glucose consumption (one-way ANOVA, **P* < 0.05, ***P* < 0.01). (A, C) HeLa cells were pretreated with 20 μM LY294002, a PI3K inhibitor, for 1 hour, then infected with *Cm* TC0668^{wt} and TC0668^{mut} strains for 24 hours. Glucose levels in the culture supernatant were quantified using the ACA assay, and GLUT1 expression was assessed

by Western blotting. (B, D) HUVECs were treated with 20 μ M LY294002 and subsequently infected with *Cm* TC0668^{wt} and TC0668^{mut} strains for 42 hours. GLUT1 levels were analyzed by Western blotting, with β -actin as the loading control.

Page 27, lines 625-629: Figure 7. Regulation of aerobic glycolysis by TC0668 of *Cm* may not rely on PI3K/AKT signaling activation (one-way ANOVA, * $P < 0.05$, **** $P < 0.0001$). (A, B) Lactate levels in the culture supernatant were measured using the ACA assay after HeLa cells (A) or HUVECs (B) were infected with *Cm* TC0668^{wt} and TC0668^{mut} strains for 24 hours and 42 hours, respectively, following a 1-hour pretreatment with LY294002 inhibitor.

Page 27, lines 630-633: Figure 8. Modulation of intracellular ATP levels by TC0668 of *Cm* in relation to PI3K/AKT signaling pathway activity (one-way ANOVA, * $P < 0.05$, ** $P < 0.01$). (A, B) After inhibiting the PI3K/AKT pathway, intracellular ATP levels were measured using a fluorescent microplate reader.

- Throughout the manuscript, the authors utilize whole-cell lysates for experimentation, as opposed to utilizing percoll-gradient enriched EBs. There is concern that the phenotypes being investigated (particularly represented by the 6hpi time point for HeLa cells) could be adversely impacted by the exposure of fresh host cells to cell debris present in whole-cell lysates. The appropriate negative control (when using EB lysates) would be lysates prepared from un-infected cells. The authors do not appear to have done this, relying instead on uninfected cells for their negative control(s).

Reply: Thank you for your valuable suggestion. We have carefully addressed each of your concerns as follow:

- 1) Our study aimed to investigate the global changes in host cell glycometabolism during infection. The use of infected cell lysates allows for a comprehensive reflection of the integrated effects of host-pathogen interactions throughout the infection process.
- 2) Uninfected cells were selected as negative controls to directly compare metabolic differences between infected and uninfected states, as they better represent the "baseline/normal" metabolic levels. Additionally, similar studies (HC et al 2002, PMID:11825770) have employed uninfected cells as negative controls to assess pathogen-induced metabolic dynamics, supporting the validity of our approach.

- It is unclear why unpaired t-tests were chosen as opposed to two-way ANOVAs (with multiple comparisons) in assessing significance for the majority of the figures that represent experimentation conducted with 2 independent variables (*Cm* strain and time). Running experiments for different time durations and only reporting 'significance by unpaired t-test for specific time points gives the impression of presenting biased comparisons.

Reply: Thank you for your rigorous critique of the statistical approach. The original manuscript mistakenly described the analysis as “unpaired t-test”. In reality, multiple t-tests (independent group comparisons at each time point) were performed. We have corrected this error in the revised manuscript and sincerely apologize for the confusion. We agree that two-way ANOVA (with multiple comparisons) is more appropriate for

multi-variable analysis, but finally we chose multiple t-tests after thinking carefully, it can be explained by the following reasons:

- 1) Samples at different time points were analyzed independently, with no longitudinal comparisons. Multiple t-tests are appropriate for independent group comparisons.
- 2) The study aimed to identify TC0668's metabolic effects at specific infection stages, not to assess interactions between time and strain variables. Multiple t-tests directly address whether strain-specific differences exist at a given time point.
- 3) With limited biological replicates (n=3 per time point), two-way ANOVA might lack statistical power, whereas multiple t-tests are more robust for small sample sizes.

- For every use of PI3K inhibitors, the authors need to demonstrate that they are not impacting the developmental cycle of their two *Cm* strains; there is a high likelihood that their observations could be indirect effects.

Reply: Thank you very much for your kind reminding. The PI3K inhibitor LY294002 used in this study is a broad-spectrum host kinase inhibitor, which primarily targets the host PI3K/AKT pathway rather than directly acting on the pathogen to influence its growth. In our data, LY294002 inhibited PI3K/AKT in TC0668^{wt}-infected cells but not in TC0668^{mut}-infected cells (Figure 5C, D), indicating pathogen-specific modulation rather than off-target effects.

We fully appreciate your emphasis on experimental rigor. The current analysis is grounded in existing data and literature. Should further validation be deemed necessary, we will prioritize this direction in subsequent studies.

Minor Comments:

- Please use page and line numbers for all subsequent submissions / revisions

Reply: Thanks a lot for your suggestion. Page and line numbers have been used.

- If using shorthand for *Chlamydia trachomatis* and *Chlamydia muridarum*, please italicize (ie. *Ct* and *Cm*)

Reply: Thank you for your attention to formatting consistency. As requested, we have revised all instances of the abbreviations *Ct* (*Chlamydia trachomatis*) and *Cm* (*Chlamydia muridarum*) to be italicized throughout the manuscript and thoroughly checked the entire text (including figure legends and table captions) to ensure compliance. Please let us know if further adjustments are needed.

- Introduction: paragraph 3, line 8: *Chlamydia muridarum*, as opposed to *Chlamydia murine*

Reply: Thank you very much for pointing out our mistake , and kindly be informed they' ve been corrected. We are very sorry for our careless mistakes. “*Chlamydia murine*” has been modified to “*Chlamydia muridarum*” in the 8th line of the 3rd paragraph of the introduction. The revision is as follow:

Page 4, line 76: have also been partially corroborated in studies of *Chlamydia muridarum* (*Cm*).

- Methods: The authors need to justify the use of human cell lines for their examination, which is primarily examining the impacts of a *Chlamydia muridarum* virulence factor. Put more simply, given that the phenotype leading to this

investigation was murine-specific, why did the authors not utilize murine cells for their study? Given that Conrad 2016 used HeLa cells to demonstrate that there was no difference in development between wt and TC0668, it would be nice to confirm this in murine cell lines.

Reply: Thanks a lot for the expert's suggestions. In fact, it's really an interesting topic to choose a cell line. Combining the aim of this study, we'd better carry out the project in mouse cells, but finally we chose HeLa and HUVEC cells after thinking carefully, it can be explained by the following reasons:

1) Though C57epi.1 or Bm1.11, cloned oviduct epithelial cell line have previously been described in Chlamydial researches (Jayarapu et al., 2009 PMID:19667042; Johnson, 2004 PMID:15213139), the cells must be grown and maintained in special cell medium with extra glutamine and growth factor supplement, which might increase the complexity and uncertainties of project.

2) HeLa and HUVEC cells have been widely used in chlamydial studies (Bulir et al., 2015 PMID:26272448; Gallegos et al., 2019, PMID:30700982; Siegl et al., 2014, PMID:25437549). Most importantly, previous study has confirmed that both murine cells BM12.4 and human cells infected with different chlamydial strains (*C. trachomatis* serovars A to H, L1 to L3, *C. muridarum*, *C. pneumoniae* and *C. caviae*) exhibited the same protein tyrosine phosphorylation patterns (Virok et al., 2005, PMID:15784533). In this respect, using human HeLa and HUVEC cells instead of

mouse cell line is practicable and facilitates comparison with a large body of published data.

3) This study focuses on the PI3K/AKT signaling pathway and glucose metabolic regulatory networks, which are highly conserved in mammals (Roy et al., 2015, PMID:26633882; Pham et al., 2020, PMID:32521639). We consider it reasonable to extrapolate the experimental results to mouse models.

We have discussed the reasons that we chose human cells instead of a mouse cell line.

This new part is as follows:

Page 13, lines 280-289: HeLa and HUVEC cells have been widely used in chlamydial studies^[15,20,21]. Most importantly, previous study has confirmed that both murine cells BM12.4 and human cells infected with different chlamydial strains (*C. trachomatis* serovars A to H, L1 to L3, *C. muridarum*, *C. pneumoniae* and *C. caviae*) exhibited the same protein tyrosine phosphorylation patterns^[22]. In addition, PI3K/AKT signaling pathway and glucose metabolic regulatory networks, which are highly conserved in mammals^[23,24]. It's practicable to apply human HeLa and HUVEC cells to investigate the role and mechanism of TC0668 in regulating cellular glucose metabolism. This work aims to provide a foundation for understanding the role of TC0668 in the pathogenic mechanisms of *Cm*.

- **Methods: Please indicate how cells were infected for each experiment (rocking, static, centrifugation), the duration of initial infections, and whether crude lysates**

were removed and replaced with fresh culture medium prior to subsequent incubation.

Reply: Thank you for your comments. We are very sorry for unclear presentation in the original manuscript and have added this part. Detailed modifications of this part are as follow:

Page 6, lines 117-123: HeLa cells were exposed to *Cm* TC0668^{wt} or TC0668^{mut} monoclonal strains at a multiplicity of infection (MOI) of 1.0, followed by centrifugation at 1000 rpm for 1 hour at 37 °C. After removal of the inoculum, the cells were replenished with fresh DMEM with 10% FCS for 6, 12, 18, or 24 hours. HUVECs were similarly infected with these strains at an MOI of 1.0, followed by centrifugation at 1000 rpm for 1 hour at 37 °C. After removal of the inoculum, the cells were replenished with fresh ECM with 10% FCS for 12, 24, 36, or 42 hours.

- paragraph 1, lines 3-5: reference(s) for this statement are requested

Reply: Thank you for your comments and helpful suggestions. We have added the references in page 18, lines 415-418. The detailed revisions of this part are as follow:

Page 19, lines 428-431: [1] Schachter J, Caldwell HD. Chlamydiae[J]. Annu Rev Microbiol, 1980, 34:285-309.

[2] Wenbo L, Yewei Y, Hui Z, et al. Hijacking host cell vesicular transport: New insights into the nutrient acquisition mechanism of Chlamydia[J]. Virulence, 2024, 15(1):2351234.

- Figure 2: The GLUT1 data looks strong, but the Glucose levels look weaker. Please indicate in figure legends exactly what type of statistical analysis was conducted and

how many replicates data presented represents.

Reply: Thank you very much for your helpful reminding of the Figure. All data were analyzed using multiple t-tests to compare differences between TC0668^{wt}- and TC0668^{mut}-infected groups. That's because the experimental design involves independent sample comparisons (different strain infections). And each condition was tested in three independent biological replicates, with data presented as mean \pm SD. Per your helpful suggestion, the figure legend now is as follow:

Page 25, lines 579-587: Figure 1. Deficiency in TC0668 impairs glucose uptake regulated by *Cm* in host cells (multiple t-tests, * $P < 0.05$, ** $P < 0.01$, *** $P < 0.001$). (A) Glucose levels in the supernatant of HeLa cells infected with *Cm* TC0668^{wt} and TC0668^{mut} strains were measured using the ACA assay at 6, 12, 18, and 24 hours post-infection. (B) Similarly, glucose concentrations in the supernatant of HUVECs infected with these strains were analyzed at 12, 24, 36, and 42 hours using ACA. (C) GLUT1 protein expression was assessed by Western blotting in HeLa cells infected with *Cm* TC0668^{wt} and TC0668^{mut} strains at 6, 12, 18, and 24 hours. (D) Western blotting was also used to evaluate GLUT1 expression in HUVECs infected with these strains at 12, 24, 36, and 42 hours. β -actin served as a loading control.

- Figure 3: From the blot shown in panel A, it certainly looks like COX IV is elevated in the WB post 18hpi. Please describe how the 'relative levels' were calculated, what the error bars represent, and how many experimental replicates were used to generate each data point.

Reply: Thank you very much for your helpful reminding of the Figure. We respond to your concerns as follows:

- 1) Grayscale analysis of Western blot images was performed using ImageJ software and values of target protein and loading control (β -actin) bands were measured. The relative expression level was calculated as the ratio of target protein to loading control (Target Grayscale / β -actin Grayscale).
- 2) Error bars represent the standard deviation (SD) derived from three independent biological replicates.
- 3) All data were independently repeated three times.

These methodological details have been added to the revised “Materials and Methods” .

Detailed modifications of this part are as follow:

Page 7, lines 148-152: Grayscale analysis of Western blot images was performed using ImageJ software and values of target protein and loading control (β -actin) bands were measured. The relative expression level was calculated as the ratio of target protein to loading control (Target Grayscale / β -actin Grayscale). Error bars represent the standard deviation (SD) derived from three independent biological replicates.

- It seems puzzling that, given the author's conclusion that TC0668 is responsible for inducing cells into a hypermetabolic state, the absence of this state would have no discernable impact on Chlamydial biology / development. How do the authors account for this?

Reply: Thank you for raising this important point. The apparent paradox can be

explained by the reason: TC0668 don't acts as a pathogen-intrinsic developmental regulator. While TC0668 induces a hypermetabolic state in host cells, its absence does not impair *C. muridarum*'s intrinsic developmental cycle (e.g., inclusion formation, RB-to-EB differentiation), as demonstrated by Conrad et al. (2016, PMID: 26597987) and Wang Y et al. (2019, PMID: 31787950) in host cells. These suggest that TC0668-mediated metabolic reprogramming creates a favorable microenvironment for pathogen survival and persistence (e.g., via enhanced ATP/nutrient availability), but is not strictly required for its basic replication.

- To the above point, given what we know about the growth of *C. muridarum* in culture, it could be that the authors utilize such 'nutrient-rich' conditions (ie. serum + human cell lines) that any impact on the host cell's metabolic state is immaterial to chlamydial development. This could be addressed with a relatively straightforward approach: remove FCS (not required for *Cm* culture) from culture medium and examine potential differences in ifu counts from HeLa cells infected with WT vs. TC0668 null strains at early (12-16hpi) and later (24 hpi) developmental time points. Similar experimentation could also be conducted between strains in a murine-derived cell line.

Reply: Thanks a lot for your comments. We respond to your concerns as follows:

1) The growth factors and nutrients in serum are essential for maintaining the normal metabolic state of host cells. Existing studies suggest that serum-free conditions may

induce stress-induced metabolic adaptations in host cells (Maria Antònia Forteza-Genestra et al., 2020, PMID:32059497; EMBO J, 2012, PMID:22728825; Rother et al., 2018, PMID; 29706504), which could confound the analysis of pathogen-specific regulatory mechanisms. We fully acknowledge the value of your suggestion for future mechanistic studies.

2) Conrad et al. (2016, PMID:26597987) and Wang Y et al. (2019, PMID: 31787950) systematically demonstrated that TC0668^{wt} and TC0668^{mut} strains exhibit comparable developmental cycles (IFU counts) in host cells, indicating that TC0668 primarily modulates host interactions (e.g., metabolism) rather than directly affecting chlamydial proliferation. While IFU data were not repeated here, we fully acknowledge the value of your suggestion for future mechanistic studies.

Reviewer #2:

In the manuscript titled “The influence of TC0668 on glycometabolism modulation in Chlamydia muridarum-infected host cells”, Yu et al continue their studies of TC0668, a gene that they and others have previously shown to be important for infection in mouse models. They demonstrated that the presence of wild-type TC0668 increases host cell expression of several genes involved in host cell metabolism. They also showed a modest drop in media glucose concentration that is exacerbated in the presence of TC0668. Wild-type TC0668 also appears to be critical for AKT phosphorylation, particularly during mid- to late-stage infection. Overall, the data appear to support the conclusion that TC0668 plays a role in host

metabolism. However, some additional discussion and further clarifications are required to complete this paper.

Reply: Thanks for your approval and encouragement. We have carefully revised the manuscript to enhance its clarity, and our changes/additions to the manuscript are given in the red text.

- The authors should consider moving figure 1 to supplemental. Alternatively, they could expand the schematic to integrate their proposed model. As is, it adds little to the paper.

Reply: Thank you for your constructive feedback on improving the manuscript's clarity. As suggested, we have moved Figure 1 (Schematic representation of cellular glucose metabolism, encompassing glycolysis and oxidative phosphorylation.) to the Supplementary Materials (Page 1, Lines 1-3), labeled as Supplementary Figure S1.

- The mechanism for the observed alterations to host cell metabolism should be discussed. The authors' lab has previously demonstrated that TC0668 is likely *Cm* membrane-bound rather than an inclusion membrane protein. Do the authors predict that TC0668 has enzymatic activity? Does TC0668 regulate deubiquitination of GLUT1, a known phenomenon (see Wang et al 2017, PMID: 29040458)? While speculative, further discussion of the hypothesized mechanism would enhance the discussion.

Reply: Thanks for your helpful suggestions. We have expanded the discussion. The description of complementary discussion is as follows:

Page 14, lines 302-305: TC0668, likely membrane-bound, may regulate host metabolism indirectly. While its enzymatic activity remains uncharacterized, Wang et al demonstrated that GLUT1 stability is regulated by deubiquitination^[25], and TC0668 may enhance GLUT1 expression through a similar mechanism.

- **Please indicate the number of biological replicates performed for each assay.**

Reply: Thanks for your kind reminding. We are very sorry for unclear presentation in the original manuscript about the number of biological replicates. All data were acquired from three independent experiments. We have added this part to the revised “Statistical Analysis”. Detailed modifications of this part are as follow:

Page 8, lines 178-179: All data were acquired from three independent experiments.

- **Please indicate the conditions of the control samples for each assay, especially the temporal experiments. Is the control taken at the end of an identical incubation period as the final timepoint?**

Reply: Thank you for your comments. We have added this part to the revised “Materials and Methods”. Detailed modifications of this part are as follow:

Page 7-8, lines 146-148、159-161、173-175: Control samples (uninfected groups) were processed synchronously with infected groups and harvested at the same endpoint of the incubation period.

- **Given the intrinsic variability of western blot densitometry, the authors should provide more detail on the statistical analyses used in each instance. If using ANOVA, what post hoc tests were used?**

Reply: Thank you very much for your helpful reminding. We are very sorry for unclear presentation in the original manuscript. These details have been added to the revised “Statistical Analysis” and captions. Detailed modifications of this part are as follow:

Page 8, lines 178-181: All data were acquired from three independent experiments. The data were analyzed using multiple t-tests and one-way ANOVA. Error bars represent the standard deviation (SD) derived from three independent biological replicates.

Page 25, lines 579-587: Figure 1. Deficiency in TC0668 impairs glucose uptake regulated by *Cm* in host cells (multiple t-tests, **P* <0.05, ***P* <0.01, ****P* <0.001). (A) Glucose levels in the supernatant of HeLa cells infected with *Cm* TC0668^{wt} and TC0668^{mut} strains were measured using the ACA assay at 6, 12, 18, and 24 hours post-infection. (B) Similarly, glucose concentrations in the supernatant of HUVECs infected with these strains were analyzed at 12, 24, 36, and 42 hours using ACA. (C) GLUT1 protein expression was assessed by Western blotting in HeLa cells infected with *Cm* TC0668^{wt} and TC0668^{mut} strains at 6, 12, 18, and 24 hours. (D) Western blotting was also used to evaluate GLUT1 expression in HUVECs infected with these strains at 12, 24, 36, and 42 hours. β -actin served as a loading control.

Page 25-26, lines 588-593: Figure 2. The effect of *Cm* TC0668 on mitochondrial oxidative phosphorylation (OXPHOS) in host cells was evaluated (multiple t-tests, **P* < 0.05). Western blot analysis was conducted to measure the expression levels of OGDH and COX IV in HeLa cells infected with *Cm* TC0668^{wt} and TC0668^{mut} strains for various durations (6, 12, 18, and 24 hours), and in HUVECs infected for different periods (12, 24,

36, and 42 hours). β -actin served as a loading control.

Page 26-27, lines 604-616: Figure 5. The TC0668 compound derived from *Cm* significantly stimulates the activation of the PI3K/AKT signaling pathway. (A) Western blot analysis was performed to assess the expression of PI3K and the phosphorylation of AKT in HeLa cells infected with the *Cm* TC0668^{wt} and TC0668^{mut} strains at 6, 12, 18, and 24 hours (multiple t-tests, * $P < 0.05$, ** $P < 0.01$, *** $P < 0.001$). (B) The presence of PI3K and p-AKT in HUVECs was examined by Western blotting following infection with the *Cm* TC0668^{wt} and TC0668^{mut} strains at 12, 24, 36, and 42 hours (multiple t-tests, * $P < 0.05$, ** $P < 0.01$, *** $P < 0.001$) (C) HeLa cells were treated with the PI3K inhibitor LY294002 at a concentration of 20 μ M and subsequently infected with the *Cm* TC0668^{wt} and TC0668^{mut} strains for 24 hours (one-way ANOVA, * $P < 0.05$, ** $P < 0.01$, *** $P < 0.001$). (D) The phosphorylation of AKT in HUVECs infected with the *Cm* TC0668^{wt} and TC0668^{mut} strains was assessed after treatment with a specific PI3K inhibitor at a concentration of 20 μ M for 42 hours (one-way ANOVA, * $P < 0.05$, ** $P < 0.01$, *** $P < 0.001$). β -actin was used as a loading control.

Page 27, lines 617-624: Figure 6. Regulation of the PI3K/AKT pathway by TC0668 of *Cm* affects glucose consumption (one-way ANOVA, * $P < 0.05$, ** $P < 0.01$). (A, C) HeLa cells were pretreated with 20 μ M LY294002, a PI3K inhibitor, for 1 hour, then infected with *Cm* TC0668^{wt} and TC0668^{mut} strains for 24 hours. Glucose levels in the culture supernatant were quantified using the ACA assay, and GLUT1 expression was assessed by Western blotting. (B, D) HUVECs were treated with 20 μ M LY294002 and

subsequently infected with *Cm* TC0668^{wt} and TC0668^{mut} strains for 42 hours. GLUT1 levels were analyzed by Western blotting, with β -actin as the loading control.

• **The authors should detail how they determine relative levels in the blot quantitation. I assume this is relative to loading control, but explicit detail is needed.**

Reply: Thanks a lot for your kind reminding. We are very sorry for unclear presentation in the original manuscript and have added this part to the revised “Materials and Methods”. Detailed modifications of this part are as follow:

Page 7, lines 148-152: Grayscale analysis of Western blot images was performed using ImageJ software and values of target protein and loading control (β -actin) bands were measured. The relative expression level was calculated as the ratio of target protein to loading control (Target Grayscale / β -actin Grayscale). Error bars represent the standard deviation (SD) derived from three independent biological replicates.

• **While not statistically significant by the tests used, the displayed western blot and presented data imply that COXIV is affected by TC0668. The authors would benefit from displaying individual data points as opposed to averages and error bars. The error bars should also be defined (SD, SEM, etc.).**

Reply: Thank you very much for your valuable suggestion. We have carefully addressed your concerns as follow:

- 1) The mean of three independent replicates (n=3) provides a robust representation of trends, minimizing bias from single-experiment variability.
- 2) Error bars represent the standard deviation (SD) derived from three independent

biological replicates.

- **Regarding the discussion section on the 14 different GLUT transporters, Wang et al 2017 assessed transcription of all 14 and found only GLUT1 and GLUT3 upregulated. Discussion of that paper could help resolve some of those questions.**

Reply: Thanks for your helpful suggestions. We have expanded the discussion. The description of complementary discussion is as follows:

Page 19, lines 422-426: In addition, Wang et al clearly demonstrated that GLUT1 and GLUT3 were significantly upregulated in Chlamydia infection^[25]. We focus on GLUT1 due to its highly conserved and widely distributed glucose transporter in mammalian cells^[55,56]. Further investigation of other GLUT subtypes, such as GLUT3 in specific contexts is needed.

Reference in replying:

Schachter J, Caldwell HD. Chlamydiae[J]. Annu Rev Microbiol, 1980, 34:285-309.

Wenbo L, Yewei Y, Hui Z, et al. Hijacking host cell vesicular transport: New insights into the nutrient acquisition mechanism of Chlamydia[J]. Virulence, 2024, 15(1):2351234

Wang X, Hybiske K, Stephens RS. Orchestration of the mammalian host cell glucose transporter proteins-1 and 3 by Chlamydia contributes to intracellular growth and infectivity[J]. Pathog Dis, 2017, 75(8).

Cho SJ, Moon JS, Nikahira K, et al. GLUT1-dependent glycolysis regulates exacerbation of fibrosis via AIM2 inflammasome activation[J]. *Thorax*, 2020, 75(3):227-236.

Lu J, Liu X, Zheng J, et al. Lin28A promotes IRF6-regulated aerobic glycolysis in glioma cells by stabilizing SNHG14[J]. *Cell Death Dis*, 2020, 11(6):447.

Gérard HC, Freise J, Wang Z, et al. Chlamydia trachomatis genes whose products are related to energy metabolism are expressed differentially in active vs. persistent infection[J]. *Microbes Infect*. 2002;4(1):13-22.

Jayarapu K, Kerr MS, Katschke A, Johnson RM. Chlamydia muridarum-specific CD4 T-cell clones recognize infected reproductive tract epithelial cells in an interferon-dependent fashion[J]. *Infect Immun*. 2009;77(10):4469-4479.

Johnson RM. Murine oviduct epithelial cell cytokine responses to Chlamydia muridarum infection include interleukin-12-p70 secretion[J]. *Infect Immun*. 2004;72(7):3951-3960.

Re: Spectrum03051-24R1 (**The influence of TC0668 on glycometabolism modulation in Chlamydia muridarum-infected host Cells**)

Dear Dr. Zhou Zhou:

Thank you for the privilege of reviewing your work. Below you will find my comments, instructions from the Spectrum editorial office, and the reviewer comments.

Revision Guidelines

Sincerely,
Simone Filardo
Editor
Microbiology Spectrum

Reviewer #1 (Comments for the Author):

The authors have attempted to address my initial concerns. They have addressed the vast majority of my minor comments; however, they have failed to adequately address the two major points below:

Proper statistical analysis:

The authors utilize 'multiple t-tests' instead of two-way ANOVAs (with multiple comparisons) for analyzing data obtained from an experimental approach focused on 2 independent variables (bacterial strain and time). In their response to comment, the authors state:

"...the study aimed to identify TC0668's metabolic effects at specific infection stages, not to assess interactions between time and strain variables. Multiple t-tests directly address whether strain-specific differences exist at a given time point..." and that "...with limited biological replicates (n=3 per time point), two-way ANOVA might lack statistical power, whereas multiple t-tests are more robust for small sample sizes."

These arguments are not convincing. To carry out their study's aim, they are in fact comparing separate strains (wt and mutant) to each other over a number of time points. The appropriate statistical test (assuming normality) is a two-way ANOVA (with multiple comparisons). If the phenotype is significant, 3 independent biological replicates should be sufficient.

Indirect effects of PI3K inhibitors:

Chlamydia species reside within eukaryotic cells. Anything that can impact the functioning of a eukaryotic cell has the potential to adversely affect chlamydial growth, replication, and development. If you use a broad-spectrum PI3K inhibitor on a cell infected with chlamydia, you run the risk of impacting the microbe's development, and thus, indirectly impacting the phenotype under observation. A number of PI3K inhibitors have been shown in the past to adversely impact the growth and development of Chlamydial species.

The authors must demonstrate that their PI3K inhibitors do not adversely impact the chlamydial developmental cycle, or at least demonstrate that they impact the development of their wt and mutant strains equally. The easiest way to do so would be to treat chlamydia-infected cells, allow the developmental cycle to proceed, harvest EBs at the end of the developmental cycle, and then show that just as many viable EBs were obtained from PI3K inhibitor-treated cells as from non-PI3K inhibited cells (ie. a standard chlamydial inclusion forming unit; IFU count). Without this standard control, the data presented is not fully interpretable.

Reviewer #2 (Comments for the Author):

For the GLUT1 ubiquitination discussion, I would recommend expanding on whether this is a diffusible molecule or some other mechanism to enhance deubiquitinase activity. Mediation of deubiquitination without directly contacting the ubiquitin moiety or the deubiquitinases is highly unusual and would be an interesting mechanism on which to follow up.

For discussion of the GLUT transporters, please move that section up to the paragraph at lines 390-401, as it does not make sense where it is.

As a final note, including first names when referencing previous studies is highly unusual (i.e., Anthony A. Azenabor et al. in line 326). I would recommend editing these instances to just the last name (i.e., Azenabor et al.).

The changes made by Yu et al address the concerns regarding experimental clarity and improve the manuscript. To my point about the discussion of GLUT1 stability and the GLUT transporters, I would encourage the authors to expand more on the discussion of these aspects.

For the GLUT1 ubiquitination discussion, I would recommend expanding on whether this is a diffusible molecule or some other mechanism to enhance deubiquitinase activity. Mediation of deubiquitination without directly contacting the ubiquitin moiety or the deubiquitinases is highly unusual and would be an interesting mechanism on which to follow up.

For discussion of the GLUT transporters, please move that section up to the paragraph at lines 390-401, as it does not make sense where it is.

As a final note, including first names when referencing previous studies is highly unusual (i.e., Anthony A. Azenabor et al. in line 326). I would recommend editing these instances to just the last name (i.e., Azenabor et al.).

Manuscript number: Spectrum03051-24

Title: The influence of TC0668 on glycometabolism modulation in Chlamydia muridarum-infected host cells

Dear Editors and Reviewers:

We thank all the editors and reviewers for their valuable comments and suggestions. We have carefully revised the manuscript to enhance its clarity, and our changes/additions to the manuscript are given in the red text. Our point to point responses are presented in the following. We hope that the revision would satisfactorily address the comments and concerns of the editors and reviewers.

Reply to the reviewer's comments:

Reviewer #1:

The authors have attempted to address my initial concerns. They have addressed the vast majority of my minor comments; however, they have failed to adequately address the two major points below:

Proper statistical analysis:

The authors utilize 'multiple t-tests' instead of two-way ANOVAs (with multiple comparisons) for analyzing data obtained from an experimental approach focused on 2 independent variables (bacterial strain and time). In their response to comment, the authors state:

"...the study aimed to identify TC0668's metabolic effects at specific infection stages, not to assess interactions between time and strain variables. Multiple t-tests directly address whether strain-specific differences exist at a given time point..." and that "...with limited biological replicates (n=3 per time point), two-way ANOVA might lack statistical power, whereas multiple t-tests are more robust for small sample sizes."

These arguments are not convincing. To carry out their study's aim, they are in fact comparing separate strains (wt and mutant) to each other over a number of time points. The appropriate statistical test (assuming normality) is a two-way ANOVA (with multiple comparisons). If the phenotype is significant, 3 independent biological replicates should be sufficient.

Reply: We thank the reviewer for pointing out the inappropriateness of using multiple t-tests instead of two-way ANOVA. We have now re-analysed all datasets that contain two independent variables (bacterial strain and time) by two-way ANOVA. Specifically:

Fig 1 to Fig 5: both data sets were re-analysed by two-way ANOVA; updated P values and significance symbols are highlighted in red in the figure legends.

These changes do not alter the biological conclusions, but they provide a statistically rigorous evaluation of strain and time effects. All modifications are clearly indicated in the revised manuscript and detailed corrections are listed as below:

Page 8, lines 179-180: The data were analyzed using two-way ANOVA and one-way ANOVA.

Page 9, lines 191-193: Notably, compared to the TC0668^{mut}-infected group, glucose levels in the culture medium dropped more dramatically following infection with the *Cm* TC0668^{wt} strain, particularly in HeLa cells at 12 and 18 h, and in HUVECs at 24, 36, and 42 h (Figure 1A, B).

Page 25, lines 586-587: **Figure 1.** Deficiency in TC0668 impairs glucose uptake regulated by *Cm* in host cells (two-way ANOVA, * $P < 0.05$, ** $P < 0.01$, *** $P < 0.001$, **** $P < 0.0001$).

Page 26, lines 596-597: **Figure 2.** The effect of *Cm* TC0668 on mitochondrial oxidative phosphorylation (OXPHOS) in host cells was evaluated (two-way ANOVA, * $P < 0.05$).

Page 26, lines 602-603: **Figure 3.** TC0668 deficiency disrupts *Cm*-regulated aerobic glycolysis in host cells (two-way ANOVA, * $P < 0.05$, ** $P < 0.01$, *** $P < 0.001$).

Page 26, lines 607-608: **Figure 4.** The absence of TC0668 affects *Cm*'s modulation of intracellular ATP levels in host cells (two-way ANOVA, * $P < 0.05$, **** $P < 0.0001$).

Page 27, lines 613-618: (A) Western blot analysis was performed to assess the expression of PI3K and the phosphorylation of AKT in HeLa cells infected with the *Cm* TC0668^{wt}

and TC0668^{mut} strains at 6, 12, 18, and 24 hours (two-way ANOVA, * $P < 0.05$, *** $P < 0.001$). (B) The presence of PI3K and p-AKT in HUVECs was examined by Western blotting following infection with the *Cm* TC0668^{wt} and TC0668^{mut} strains at 12, 24, 36, and 42 hours (two-way ANOVA, * $P < 0.05$, ** $P < 0.01$, *** $P < 0.001$)

Indirect effects of PI3K inhibitors:

Chlamydia species reside within eukaryotic cells. Anything that can impact the functioning of a eukaryotic cell has the potential to adversely affect chlamydial growth, replication, and development. If you use a broad-spectrum PI3K inhibitor on a cell infected with chlamydia, you run the risk of impacting the microbe's development, and thus, indirectly impacting the phenotype under observation. A number of PI3K inhibitors have been shown in the past to adversely impact the growth and development of Chlamydial species.

The authors must demonstrate that their PI3K inhibitors do not adversely impact the chlamydial developmental cycle, or at least demonstrate that they impact the development of their wt and mutant strains equally. The easiest way to do so would be to treat chlamydia-infected cells, allow the developmental cycle to proceed, harvest EBs at the end of the developmental cycle, and then show that just as many viable EBs were obtained from PI3K inhibitor-treated cells as from non-PI3K

inhibited cells (ie. a standard chlamydial inclusion forming unit; IFU count).

Without this standard control, the data presented is not fully interpretable.

Reply: We appreciate the reviewer's concern that PI3K inhibitors might non-specifically impair the chlamydial developmental cycle. To rigorously exclude this possibility, we infected both HeLa and HUVEC cells with either the *Cm* wild-type and the *Cm* TC0668^{mut} strain and added the same concentration of the PI3K inhibitor used in our main experiments. After allowing a complete developmental cycle, we harvested elementary bodies (EBs) and quantified viable progeny by standard IFU assays. Consistent with this control experiment, the inhibitor had no adverse effect on chlamydial growth or development; thus, any phenotypic differences between the *Cm* wild-type and the *Cm* TC0668^{mut} strain can be attributed specifically to the TC0668 mutation rather than to off-target effects of the inhibitor.

We have included the relevant results in Supplementary Fig.2 and added the corresponding conclusions to the main text, as detailed below:

Page 11, lines 251-252: We have demonstrated that the inhibitor exerts no effect on the growth or development of either chlamydial strain (Supplementary Fig. 2).

Reviewer #2:

For the GLUT1 ubiquitination discussion, I would recommend expanding on whether this is a diffusible molecule or some other mechanism to enhance deubiquitinase activity. Mediation of deubiquitination without directly contacting

the ubiquitin moiety or the deubiquitinases is highly unusual and would be an interesting mechanism on which to follow up.

Reply: Thanks for your kind reminding. We have expanded the discussion. The description of complementary discussion is as follows:

Page 14, lines 303-309: TC0668, likely membrane-bound, may regulate host metabolism indirectly. While its enzymatic activity remains uncharacterized, given the important role of deubiquitination in regulating GLUT1 stability^[25] and the fact that AKT phosphorylates USP4 to enhance its stability^[26], we propose that TC0668 may sustain GLUT1 expression indirectly through the PI3K/AKT-USP4 axis. Specifically, TC0668-induced AKT activation could phosphorylate and stabilise deubiquitinases (e.g. USP4), thereby reducing GLUT1 ubiquitination and subsequent proteasomal degradation.

For discussion of the GLUT transporters, please move that section up to the paragraph at lines 390-401, as it does not make sense where it is.

Reply: Thank you for your comments. We have made changes to this part. Detailed modifications are as follow:

Page 18, lines 397-401: Wang et al clearly demonstrated that GLUT1 and GLUT3 were significantly upregulated in *Chlamydia* infection^[25]. We focus on GLUT1 due to its highly conserved and widely distributed glucose transporter in mammalian cells^[51,52]. Further investigation of other GLUT subtypes, such as GLUT3 in specific contexts is needed.

As a final note, including first names when referencing previous studies is highly unusual (i.e., Anthony A. Azenabor et al. in line 326). I would recommend editing these instances to just the last name (i.e., Azenabor et al.).

Reply: Thank you very much for your helpful reminding. We have implemented the following corrections:

Page 15, lines 330: Previous research by Azenabor et al. suggested that *Chlamydia pneumoniae* (*Cpn*) may enhance cytochrome C oxidase activity in macrophages^[31].

Reference in replying:

Zhang L, Zhou F, Drabsch Y, et al. USP4 is regulated by AKT phosphorylation and directly deubiquitylates TGF- β type I receptor[J]. Nat Cell Biol. 2012;14(7):717-726.

Re: Spectrum03051-24R2 (**The influence of TC0668 on glycometabolism modulation in Chlamydia muridarum-infected host Cells**)

Dear Dr. Zhou Zhou:

Your manuscript has been accepted, and I am forwarding it to the ASM production staff for publication. Your paper will first be checked to make sure all elements meet the technical requirements. ASM staff will contact you if anything needs to be revised before copyediting and production can begin. Otherwise, you will be notified when your proofs are ready to be viewed.

Sincerely,
Simone Filardo
Editor
Microbiology Spectrum

Reviewer #1 (Comments for the Author):

The authors have now fully addressed my initial concerns. I have no further comments.